# ADPretrain: Advancing Industrial Anomaly Detection via Anomaly Representation Pretraining

**Xincheng Yao[1], Yan Luo[3,4*], Zefeng Qian[1], Chongyang Zhang[1,2*]**
[1]School of Information Science and Electronic Engineering, Shanghai Jiao Tong University
[2]MoE Key Lab of Artificial Intelligence, AI Institute, Shanghai Jiao Tong University
[3]College of Artificial Intelligence, Nanjing Agricultural University
[4]Key Laboratory of Livestock Farming Equipment, Ministry of Agriculture and Rural Affairs,
Nanjing Agricultural University
{i-Dover, zefeng_qian, sunny_zhang}@sjtu.edu.cn[1]
luoyan@njau.edu.cn[3]

## Abstract

The current mainstream and state-of-the-art anomaly detection (AD) methods are substantially established on pretrained feature networks yielded by ImageNet pretraining. However, regardless of supervised or self-supervised pretraining, the pretraining process on ImageNet does not match the goal of anomaly detection (*i.e.*, pretraining in natural images doesn't aim to distinguish between normal and abnormal). Moreover, natural images and industrial image data in AD scenarios typically have the distribution shift. The two issues can cause ImageNet-pretrained features to be suboptimal for AD tasks. To further promote the development of the AD field, pretrained representations specially for AD tasks are eager and very valuable. To this end, we propose a novel AD representation learning framework specially designed for learning robust and discriminative pretrained representations for industrial anomaly detection. Specifically, closely surrounding the goal of anomaly detection (*i.e.*, focus on discrepancies between normals and anomalies), we propose angle- and norm-oriented contrastive losses to maximize the angle size and norm difference between normal and abnormal features simultaneously. To avoid the distribution shift from natural images to AD images, our pretraining is performed on a large-scale AD dataset, RealIAD. To further alleviate the potential shift between pretraining data and downstream AD datasets, we learn the pretrained AD representations based on the class-generalizable representation, residual features. For evaluation, based on five embedding-based AD methods, we simply replace their original features with our pretrained representations. Extensive experiments on five AD datasets and five backbones consistently show the superiority of our pretrained features. The code is available at https://github.com/xcyao00/ADPretrain.

## 1 Introduction

In recent years, the anomaly detection (AD) field has undergone rapid evolution. Researchers have proposed various deep AD methods, including reconstruction-based [5, 51, 50, 48, 16], feature-distance-based [8, 30, 33, 13], one-class-classification-based (OCC) [36, 24, 49, 23], distillation-based [3, 37, 9, 53], and generative-model-based AD methods [12, 20, 14, 52, 10], etc. Although different taxonomies of methods have their distinctive insights and collaboratively address the challenging issues in anomaly detection from various perspectives, the essence of well-performed AD methods

---

*Both Yan Luo and Chongyang Zhang are Corresponding Author.

39th Conference on Neural Information Processing Systems (NeurIPS 2025).

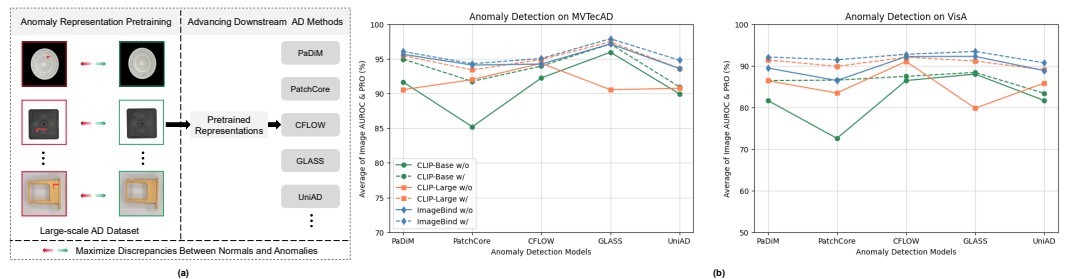

Figure 1: (a) Conceptual illustration of anomaly representation pretraining. (b) Performance comparison on MVTecAD (left) and VisA (right). "w/o" and "w/" refer to without and with our pretrained features. Under multiple AD methods and backbones, our pretrained features are consistently superior to the original features (dashed lines are overall on top of solid lines).

can still be attributed to the representation ability of features, *i.e.*, if normal and abnormal features are highly discriminative (*e.g.*, linearly separable), anomalies can be easily detected out as outliers.

Due to the scarcity and uncertainty of anomalies, the main paradigm in AD is unsupervised (with only normal samples for training). This paradigm makes it hard to learn highly discriminative representations for two reasons: (1) With only normal samples, classical AD methods resort to self-supervised proxy tasks (*e.g.*, auto-encoding reconstruction) to learn normal representations, assuming that models would fail in abnormal image regions. However, only learning normal samples may easily lead to the phenomenon of "pattern collapse" [50, 36], where both normal and abnormal features are similar. *E.g.*, anomalies may be well reconstructed in reconstruction-based methods. (2) The scale of conventional AD datasets is usually not too large, relying on limited normal training data for representation learning would also restrict the quality of learned representations.

To overcome the weaknesses of self-supervised representation learning from scratch, previous works [1, 8, 33] (starting with [1]) have confirmed that utilizing pretrained features yielded by ImageNet pretraining can significantly improve AD performance compared to features learned from scratch on the AD datasets. Afterward, mainstream and state-of-the-art AD methods are almost based on ImageNet-pretrained[2] feature networks. However, ImageNet-pretrained features are still suboptimal for AD tasks for two reasons: (1) The conventional pretraining ways (*e.g.*, image classification, image contrastive learning, masked image modeling) don't meet the goal of anomaly detection, as there is no concept of normal and abnormal during the ImageNet pretraining process. (2) Natural images in ImageNet and image data in AD scenarios typically have the remarkable distribution shift, without further adaptation, the performance of pretrained features on AD data may be constrained [39]. Therefore, to further advance the development of anomaly detection, it's necessary to construct and learn pretrained representations specially for anomaly detection tasks.

In this paper, we explore the problem of anomaly representation pretraining, where the objective is to learn pretrained AD representations that are better than ImageNet-pretrained features when applied to downstream AD methods (see Fig.1). Like conventional pretraining works, we expect that anomaly representation pretraining would be performed on a large-scale AD dataset. Luckily, the proposal of the RealIAD dataset [44] provides us with such a prerequisite. RealIAD has a large enough data scale, containing a total of 151050 images, of which 99721 images are normal and 51329 images are abnormal. Based on RealIAD, we propose a novel AD representation learning framework specially designed for learning robust and discriminative pretrained representations for industrial anomaly detection. By pretraining on RealIAD, we can avoid the distribution shift from natural images to industrial AD images. As the main characteristic of AD tasks is to focus on the discrepancies between normals and anomalies, contrastive learning should be the most suitable pretraining paradigm. For better representation learning, we propose angle- and norm-oriented contrastive losses to maximize the angle size and norm difference between normal and abnormal features simultaneously. As from the feature similarity perspective, differences between two features are reflected in angle and norm. Further considering the potential distribution shift caused by different product categories between RealIAD and downstream AD datasets, we learn the pretrained AD representations based on the

---

[2]The "ImageNet-pretrained" is to refer to feature networks pretrained on all sorts of large-scale datasets (not just ImageNet). As ImageNet is the most well-known pretraining dataset, we still use "ImageNet-pretrained".

generalizable representation in anomaly detection, residual features [46]. We find that when used as pretrained AD representations, residual features are better than vanilla features extracted by the feature network. Our contributions are as follows:

1. To the best of our knowledge, this is the first study dedicated to anomaly representation pretraining. We construct an AD representation learning framework, which can learn robust and discriminative pretrained features specially for anomaly detection tasks.

2. To fully optimize the discrepancies between normal and abnormal features, we propose angle- and norm-oriented contrastive losses, which can maximize the angle size and norm difference between two features simultaneously.

3. In five embedding-based[3] AD methods, we replace their original features with our pretrained features. Extensive experiments show that our pretrained features can consistently surpass the original features. Moreover, another merit of our pretrained features is that they are also good representations for few-shot anomaly detection by simply using feature norms as anomaly scores.

## 2 Related Work

**Anomaly Detection With Pretrained Features.** The current mainstream and state-of-the-art AD methods are mostly based on ImageNet-pretrained feature networks. Starting with [1], the authors found that simply combining pretrained features with the KNN algorithm can significantly outperform previous self-supervised methods. Afterward, ImageNet-pretrained features are widely adopted by AD methods (we call them embedding-based AD methods). Representative methods include: PaDiM and PatchCore. PaDiM [8] extracts pretrained features to model Multivariate Gaussian distribution and then utilizes the Mahalanobis distance to measure anomaly scores. PatchCore [33] proposes to utilize locally aggregated features and introduces a maximally representative memory bank of normal features. However, images in AD scenarios generally have an obvious distribution shift with natural images. To better account for the distribution shift, subsequent methods mainly follow a standard paradigm: fixing pretrained feature networks and designing learnable modules. Representative methods include: embedding-based reconstruction methods [45, 50, 16], distillation-based methods [3, 37, 9, 41, 53], and normalizing-flow-based methods [34, 12, 35, 47].

**Feature Adaptation to AD Tasks.** Due to the distribution shift, pretrained features are still suboptimal for AD tasks. To better utilize pretrained features, this line of work aims to adapt pretrained features to the target distribution in AD datasets. However, this doesn't mean simple fine-tuning is feasible, as naive fine-tuning in AD datasets often results in catastrophic collapse (feature deterioration) and reduced performance [27, 30]. To this end, PANDA [30] proposes techniques based on early stopping and EWC [22], a continual learning method, to mitigate the catastrophic collapse. In [31], Reiss, et al. propose a contrastive method, mean-shifted contrastive loss, which is more suitable for AD tasks by ensuring the compactness of normal features after fine-tuning. In FYD [54], the authors propose a dense non-contrastive learning method for self-supervised learning of compact normal features and utilize a stop-gradient strategy to alleviate the collapse. In [32], Rippel, et al. propose a Gaussian fine-tuning method, where the pretrained network is fine-tuned by minimizing the Mahalanobis distance to the estimated Gaussian distribution. However, these methods are still not general, as their fine-tuned feature networks are applicable only to the trained dataset. By comparison, our work takes a step forward beyond fine-tuning, which is dedicated to learning robust and discriminative pretrained features for downstream AD tasks. More discussions are in Appendix B.1.

## 3 Method

**Overview.** Our proposed anomaly representation learning framework is illustrated in Fig.2. Our pretrained AD representations are based on residual features. During pretraining, the residual features are transformed into a latent space by the Feature Projector (Sec.3.3), and then optimized by the angle- and norm-oriented contrastive losses (Sec.3.2) to maximize the angle size and norm difference between normal and abnormal features simultaneously. As our method aims to learn better representations for AD tasks, its usage is the same as conventional feature extractors. We can simply

---

[3]In this paper, "embedding-based" is a broad concept. We use embedding-based AD methods to refer to all sorts of AD methods that utilize pretrained feature networks.

incorporate our pretrained features outputted by the Feature Projector into existing embedding-based AD methods to replace the original features.

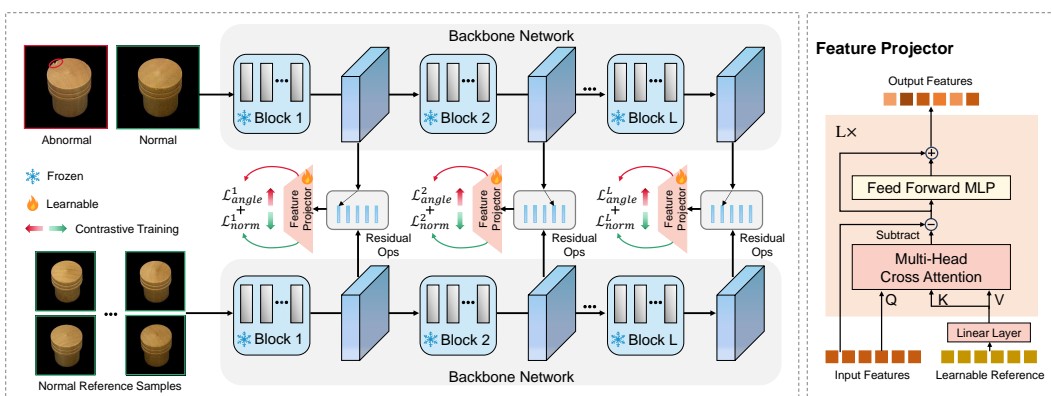

Figure 2: Framework overview. We learn pretrained AD representations based on residual features, while not the features directly produced by the backbone network. Residual features are generated by subtracting normal reference features (Sec.3.1), which are extracted from the normal reference samples. Features yielded by the Feature Projector are optimized by the angle- and norm-oriented contrastive losses (Sec.3.2). The Feature Projector is based on the Transformer architecture, but we alter self-attention to our proposed learnable key/value attention (Sec.3.3).

## 3.1 Construction of Fundamental Anomaly Detection Representations

We first consider what kind of representation features are suitable as pretrained AD representations. We expect that pretrained features can serve as fundamental (general) features in anomaly detection (*i.e.*, can perform well on various AD datasets). Therefore, we think that it's best for pretrained representations to be domain-invariant, namely, even if the data distributions in downstream AD datasets differ from that in the pretraining AD dataset, the normal and abnormal representation distributions are consistent (relatively invariant) with those during pretraining. To this end, we utilize the recently proposed class-generalizable representations in anomaly detection, residual features [46], as the fundamental AD representations and learn pretrained features. As the authors explained and verified in [46] that residual features can be regarded as class-invariant representations compared to the features directly produced by the feature extractor (please refer to [46] for more explanations), we also feel that residual features are potential to be general representations in anomaly detection.

Specifically, residual features are obtained by matching and then subtracting. For an input image $I \in \mathbb{R}^{H \times W \times 3}$, a backbone network is utilized to extract multi-level (*i.e.*, L) features. For each feature $x_{h,w}^l \in \mathbb{R}^{C_l}$ at level $l$ and position $(h, w)$, we will match it with the nearest normal reference feature $x_n^* = \mathrm{argmin}_{x \in \mathcal{R}_l} ||x_{h,w}^l - x||_2$ from the corresponding reference feature bank. The $\mathcal{R}_l = \{x_{h,w}^{l,i} | h/w/l/i \in \{1, \dots, H_l/W_l/L/K\}\}$ is the feature bank for $l$-th level, where $i$ denotes the $i$-th normal sample and $K$ is the number of normal reference samples. The residual representation of $x_{h,w}^l$ is defined as: $r_{h,w}^l = x_{h,w}^l - x_n^*$. During pretraining, for each training sample, we randomly select reference samples from the normal set to increase the diversity of residual features.

**Discussions with ResAD.** Compared to ResAD, we mainly take advantage of the residual features proposed in ResAD. However, our work and ResAD are clearly two different works, with differences in motivations, concerned problems, and implementation methods. Due to page limitation, we provide detailed difference discussions between our work and ResAD in Appendix B.4.

## 3.2 Contrastive Losses for Anomaly Representation Pretraining

We know that one main characteristic of AD tasks is the focus on discrepancies between normals and anomalies. Thus, good representation features for anomaly detection should be able to ensure sufficient discrepancies between normal and abnormal features. Naturally, to learn such representations, contrastive learning should be the most suitable pretraining paradigm (constructing normal-abnormal

contrastive pairs). From the feature similarity perspective, the discrepancies between two features are embodied in two aspects: the angle size between two features and the feature norm difference (*i.e*, mathematically, a feature is a vector, differences between two vectors are reflected in angle and norm). Therefore, we propose angle- and norm-oriented contrastive losses to maximize the angle size and norm difference between normal and abnormal features simultaneously. In the following losses, we use $x$ to represent residual features instead of $r$, as $x$ is most commonly used to represent features.

**Angle-Oriented Contrastive Loss.** This loss follows the classical contrastive loss, called InfoNCE [42] loss. Following common practice [15, 7], in each training step, we randomly sample a mini-batch of B images and randomly augment each image to obtain an augmented image, resulting in 2B data samples (*i.e.*, for $i$-th sample, the index of its augmented sample is denoted as $i' = i + B$). In this work, we sequentially apply three simple augmentations: *random color jitters*, *random gray scale*, and *random Gaussian blur*. Then, we employ the backbone network to extract features for the 2B samples, with a total of 2N[4] features. The $i$-th ($i \in \{1, 2, \dots, N\}$) feature $x_i$ and the feature from the same position in the augmented image, denoted as $x_{i'}, i' = i + N$, will be treated as a positive pair. The other $2(N-1)$ features within the mini-batch are treated as negative examples. Then, the InfoNCE loss for a positive pair $(x_i, x_{i'})$ is defined as:

$$\mathcal{L}_{nce}(x_i, x_{i'}) = -\log \left( \frac{\exp(\text{sim}(x_i, x_{i'})/\tau)}{\sum_{k=1}^{2N} \mathbb{I}_{[k \neq i]} \cdot \exp(\text{sim}(x_i, x_k)/\tau)} \right) \quad (1)$$

where $\mathbb{I}_{[k \neq i]} \in \{0, 1\}$ is an indicator function evaluating to 1 iff $k \neq i$, $\tau$ denotes a temperature parameter, and $\text{sim}(x_i, x_{i'}) = x_i^T x_{i'}/||x_i||_2 ||x_{i'}||_2$ is the cosine similarity between $x_i$ and $x_{i'}$. However, the conventional contrastive learning objective is not suitable for anomaly representation pretraining, as negative examples contain both normal and abnormal features. When $x_i$ belongs to normal features, only abnormal features should be treated as negative examples to enlarge the discrepancies between normal and abnormal features. In addition, in mathematics, features can be viewed as vectors starting from the origin. Cosine similarity in fact measures the angle between two features with the origin as the center. However, previous work [31] indicated that the standard contrastive loss prefers to optimize features uniformly distributed on the origin-centered sphere, but it is not conducive to making normal and abnormal features more discriminative. This is harmful for anomaly detection. Inspired by [31], we measure the angle between two features with respect to the center of the normal features rather than the origin. Let us denote the center of all normal features in the training set by $c = \mathbb{E}_{x \in \mathcal{X}_{normal}}[x]$. Then, for each feature $x_i$, we first subtract the center $c$ to form a center-shifted feature $\bar{x}_i = x_i - c$ with the center $c$ as origin and then calculate the cosine similarity $\text{sim}(\bar{x}_i, \bar{x}_{i'})$. In this way, the contrastive loss preserves features' distances to the center $c$ while maximizing the angles between normal and abnormal features. For a positive pair $(x_i, x_{i'})$, the angle-oriented contrastive loss is defined as:

$$\mathcal{L}_{angle}(x_i, x_{i'}) = -\log \left( \frac{\exp(\text{sim}(\bar{x}_i, \bar{x}_{i'})/\tau)}{\sum_{k=1}^{2N} \mathbb{I}_{[k \neq i]} \cdot \mathbb{I}_{[m_k \neq m_i]} \cdot \exp(\text{sim}(\bar{x}_i, \bar{x}_k)/\tau)} \right) \quad (2)$$

where $m_i \in \{0, 1\}$ is the feature label, $m_i = 0$ denotes $x_i$ is a normal feature and $m_i = 1$ denotes $x_i$ is abnormal ($m_k$ has the same meaning). In RealIAD, each sample has a ground-truth mask, we can downsample the mask to get feature labels. Further, we can analyze that the loss in Eq.(2) only uses $x_{i'}$ as the positive pair. Other features with different labels from $x_i$ are used as negative pairs. This ensures that we only perform contrast between normal and abnormal. In addition, we only utilize $x_i$ as the anchor feature, without using $x_{i'}$ as the anchor feature. That is to say, there will be no contrast between the augmented image and abnormal data in our angle-oriented contrastive loss.

**Norm-Oriented Contrastive Loss.** Unlike the above angle-oriented contrastive loss, this loss aims to enlarge the feature norm difference between normal and abnormal features. The basic idea follows one-class-classification (OCC) learning [40, 36], where normal features are optimized to be located inside the origin-centered hypersphere. In mathematics, feature norm can be viewed as the distance from the feature vector to the origin. Specifically, we use the pseudo-Huber distance $\sqrt{||x_i||_2^2 + 1} - 1$ as the feature norm, which is a more robust distance measure that interpolates from quadratic to linear penalization [24]. Then, we denote the distance from the feature $x_i$ to the hypersphere as $d_i = n_i - r$, where $n_i$ refers to the feature norm $\sqrt{||x_i||_2^2 + 1} - 1$, and $r$ is the radius of the hypersphere and is

---

[4]For symbol simplicity, we use N to represent the number of features in one feature level. Please note that we extract multi-level features, with contrastive learning on each level.

set to 0.4 (*i.e.*, when $||x_i||_2^2 = 1$, $\sqrt{||x_i||_2^2 + 1} - 1 \approx 0.4$. Setting $r$ to 0.4 is equivalent to the unit hypersphere based on Euclidean distance). The ideal learning objective is to make all normal features contracted inside the hypersphere, and for features that are already within the hypersphere, we don't need to excessively shrink towards the origin to avoid mode collapse. To this end, we propose the following contraction loss:

$$\mathcal{L}_{con}(x_i) = -\text{logsig}(-d_i) \cdot \text{e}^{d_i} \tag{3}$$

where logsig presents logarithmic sigmoid function. To better understand this loss, we further derive the gradients with respect to the model weights (denoted as $\mathcal{W}$) as follows:

$$\frac{\partial \mathcal{L}_{con}(x_i)}{\partial \mathcal{W}} = (1 - \sigma_i)\frac{\partial d_i}{\partial \mathcal{W}} - \log\sigma_i \cdot \text{e}^{d_i} \cdot \frac{\partial d_i}{\partial \mathcal{W}} \tag{4}$$

where $\sigma_i$ presents $\text{sigmoid}(-d_i)$ and can be explained as the probability that the feature inside the hypersphere. Thus, the features close to the origin have large probabilities (*i.e.*, $d_i \to -r$, $\sigma_i \to 1$) and thus small $1 - \sigma_i$ and $-\log\sigma_i$, while features outside the hypersphere have small $\sigma_i \to 0$ and large $-\log\sigma_i$ (thus large gradient values). Thus, the loss in Eq.(3) can adaptively tune attention to assign larger gradients to the features outside the hypersphere for better contraction.

For abnormal features, we expect that they are located outside the hypersphere and are highly discriminative from normal features. Thus, unlike OCC learning, we contrastively optimize the norms of abnormal features to form a certain gap between them and normal features. We further introduce a margin $\Delta r$ and define an abnormal radius $r' = r + \Delta r$. When an abnormal feature $x_j$ is inside the hypersphere with a radius $r'$, it will be pushed outside of the hypersphere. In addition, to avoid overfitting to anomalies during pretraining, we don't further push away the abnormal features that are already outside the hypersphere. Then the learning objective for $x_j$ is as follows:

$$\mathcal{L}'_{con}(x_j) = \begin{cases} -\text{logsig}(-(r' - n_j)) \cdot \text{e}^{r' - n_j}, & n_j \leq r', \\ 0, & n_j > r'. \end{cases} \tag{5}$$

Eq.(5) is based on Eq.(3), but $d_j = r' - n_j$ is reversed ($d_i = n_i - r$). We can analyze that the learning objectives in Eq.(3) and Eq.(5) are encouraging to contract normal features inside the hypersphere with radius $r$ and push abnormal features outside the hypersphere with radius $r'$. Thus, combining the two learning objectives, we call the final loss as norm-oriented contrastive loss, which is defined as:

$$\mathcal{L}_{norm}(x_i) = \mathbb{I}_{[m_i=0]} \cdot \mathcal{L}_{con}(x_i) + \mathbb{I}_{[m_i=1]} \cdot \mathcal{L}'_{con}(x_i) \tag{6}$$

In Eq.(6), we unify the symbols, $x_i$ can be either normal or abnormal, indicated by $m_i = 0$ or 1.

**Total Loss.** By combining angle- and norm-oriented contrastive losses, the whole loss for training is as follows:

$$\mathcal{L} = \frac{1}{2N}\sum_{i=1}^{2N}\lambda \cdot \mathbb{I}_{[i\leq N]} \cdot \mathcal{L}_{angle}(x_i) + \mathcal{L}_{norm}(x_i) \tag{7}$$

where $\lambda$ is set to 1 by default (please see the results in Tab.10 in Appendix E).

### 3.3 Feature Projector

As illustrated in the right part of Fig.2, the architecture of the Feature Projector is based on the popular Transformer network [43]. Differently, we replace self-attention in vanilla Transformer with our proposed Learnable Key/Value Attention. As we find that directly using vanilla Transformer as the Feature Projector has poor performance (see Tab.2(b)).

**Learnable Key/Value Attention.** Specifically, we randomly initialize $N_r$ learnable reference feature representations and define them as $\mathcal{R} \in \mathbb{R}^{N_r \times C}$. Then we utilize a fully connected layer to project $\mathcal{R}$ to the hidden dimension in Transformer into $\mathcal{R}_h \in \mathbb{R}^{N_r \times C_h}$, and a single $\mathcal{R}_h$ is used for all layers. Our idea for enhancing the vanilla self-attention is to treat input features as Query vectors and learnable reference representations as Key and Value vectors, and then apply a cross-attention between them. After cross-attention, different from the original addition operation in the residual connection, we merge the input features and attention outputs through a subtraction operation. We can understand the learnable key/value attention module in this way: the learnable reference representations can adaptively learn to represent normal patterns. As Value vectors, this makes attention outputs mainly contain normal patterns. By subtraction, we aim to eliminate the normal representations in the residual feature distribution adaptively learned by the network, further increasing the discrepancy between normal and abnormal residual features. In Sec.4.3, we provide more explanations to our Learnable Key/Value Attention based on the results in Tab.2(b).

# 4 Experiments

## 4.1 Experimental Setup

Table 1: Anomaly detection and localization results with the original features and our pretrained features. ·/· means image-level AUROC and PRO. † means our pretrained features are utilized in these AD models. For GLASS, we use the GLASS-h variant in [6].

| Model | Datasets | PaDiM [8] | PaDiM† | PatchCore [33] | PatchCore† | CFLOW [12] | CFLOW† | GLASS [6] | GLASS† | UniAD [50] | UniAD† | FeatureNorm | FeatureNorm† |
|---|---|---|---|---|---|---|---|---|---|---|---|---|---|
| **DINOv2-Base [26]** | MVTecAD | 95.6/93.1 | 95.9+0.3/92.5-0.6 | 95.5/82.7 | 99.0+3.5/87.4+4.7 | 97.7/92.3 | 98.3+0.6/92.9+0.6 | 98.3/93.5 | 99.0+0.7/95.2+1.7 | 71.1/81.5 | 97.1+26.0/91.2+9.7 | 48.4/28.9 | 98.2/92.8 |
| | VisA | 91.7/84.4 | 93.1+1.4/85.7+1.3 | 82.8/69.9 | 92.9+10.1/81.3+11.4 | 94.3/89.4 | 95.2+0.9/88.6-0.8 | 93.3/90.1 | 93.5+0.2/87.7-2.4 | 90.6/84.4 | 94.4+3.8/87.3+2.9 | 52.2/30.1 | 94.8/87.2 |
| | BTAD | 96.6/74.4 | 95.2-1.4/74.7+0.3 | 90.2/61.7 | 94.3+4.1/62.2+0.5 | 93.2/70.4 | 95.2+2.0/72.3+1.9 | 92.3/78.2 | 95.2+2.9/81.6+3.4 | 78.0/67.9 | 93.4+15.4/71.1+3.2 | 54.9/15.3 | 95.1/71.7 |
| | MVTec3D | 82.1/92.0 | 81.5-0.6/92.4+0.4 | 72.9/78.9 | 82.8+9.9/87.5+8.6 | 88.6/91.9 | 88.5-0.1/92.5+0.6 | 84.5/90.2 | 87.3+2.8/90.7+0.5 | 66.6/78.3 | 85.3+18.7/91.8+13.5 | 49.0/54.6 | 84.8/91.2 |
| | MPDD | 80.9/87.8 | 88.3+7.4/92.1+4.3 | 89.4/72.2 | 92.4+3.0/88.7+16.5 | 91.3/93.2 | 91.7+0.4/93.7+0.5 | 91.1/90.6 | 95.7+4.6/92.3+1.7 | 76.9/53.4 | 88.6+11.7/91.9+38.5 | 44.0/36.3 | 93.6/93.1 |
| **DINOv2-Large [26]** | MVTecAD | 98.7/91.0 | 98.6-0.1/92.4+1.4 | 97.6/83.8 | 99.0+1.4/88.0+4.2 | 98.8/92.7 | 98.9+0.1/93.2+0.5 | 98.4/95.3 | 99.1+0.7/96.2+0.9 | 79.6/83.0 | 96.9+17.3/91.6+8.6 | 48.4/31.4 | 98.7/93.1 |
| | VisA | 92.6/85.6 | 95.1+2.5/86.7+1.1 | 85.1/71.1 | 91.5+6.4/82.5+11.4 | 96.2/90.0 | 96.9+0.7/90.6+0.6 | 93.3/90.4 | 94.0+0.7/91.8+1.4 | 90.9/84.0 | 94.8+3.9/88.7+4.7 | 45.7/33.4 | 96.2/88.3 |
| | BTAD | 94.0/75.2 | 94.6+0.0/75.6+0.4 | 93.6/63.2 | 93.5-0.1/61.8-1.4 | 94.8/74.7 | 95.8+1.0/76.4+2.0 | 93.8/80.9 | 95.6+1.8/82.3+1.4 | 85.1/71.9 | 92.3+7.2/72.5+0.6 | 46.1/18.7 | 94.3/73.7 |
| | MVTec3D | 83.0/87.7 | 86.6+3.6/88.8+1.1 | 75.7/74.7 | 82.5+6.8/85.3+10.6 | 91.8/93.0 | 91.1-0.7/93.3+0.3 | 83.4/92.0 | 87.8+4.4/93.3+1.3 | 79.2/87.9 | 83.0+3.8/91.8+3.9 | 47.4/53.5 | 86.0/91.9 |
| | MPDD | 87.5/86.8 | 94.3+6.8/90.1+3.3 | 93.6/79.5 | 90.7-2.9/87.0+7.5 | 95.3/94.0 | 93.4-1.9/94.1+0.1 | 91.6/95.7 | 95.3+3.7/98.0+2.3 | 76.0/62.9 | 90.6+14.6/92.6+29.7 | 39.1/45.8 | 94.6/95.0 |
| **CLIP-Base [29]** | MVTecAD | 93.4/89.9 | 98.1+4.7/91.8+1.9 | 92.9/76.0 | 98.3+5.4/85.7+9.7 | 94.2/90.3 | 96.7+2.5/91.2+0.9 | 97.1/94.8 | 99.0+0.9/93.8-1.0 | 92.2/87.7 | 93.1+0.9/88.8+1.1 | 48.7/36.3 | 96.9/91.4 |
| | VisA | 87.7/75.6 | 92.6+4.9/80.3+4.7 | 81.2/63.3 | 92.5+11.3/80.6+17.3 | 89.5/83.4 | 91.5+2.0/84.7+1.3 | 92.1/85.0 | 91.4-0.7/85.4+0.4 | 90.8/79.8 | 87.0+3.5/79.8+0.0 | 48.9/31.9 | 92.5/82.6 |
| | BTAD | 93.9/70.2 | 95.4+1.5/73.2+3.0 | 91.4/58.4 | 91.6+0.2/63.6+5.2 | 94.3/71.0 | 93.9-0.4/72.7+1.7 | 92.5/80.5 | 94.3+1.8/81.3+0.8 | 90.8/66.8 | 94.2+3.4/72.5+5.7 | 47.7/14.0 | 94.1/72.0 |
| | MVTec3D | 71.7/80.8 | 81.3+9.6/87.1+6.3 | 65.0/62.7 | 77.9+12.9/84.6+21.9 | 82.1/90.1 | 83.8+1.7/90.4+0.3 | 79.2/90.8 | 80.7+1.5/89.2-0.6 | 65.3/81.6 | 82.6+17.3/90.7+9.1 | 50.9/10.4 | 81.4/90.6 |
| | MPDD | 85.6/79.3 | 92.0+6.4/89.5+10.2 | 80.5/59.9 | 90.6+10.1/88.6+28.7 | 85.2/90.4 | 90.1+1.9/92.7+2.3 | 92.6/94.2 | 94.3+1.7/94.3+0.1 | 82.2/78.4 | 86.8+4.6/91.5+13.1 | 50.8/20.6 | 93.6/91.7 |
| **CLIP-Large [29]** | MVTecAD | 89.4/91.7 | 98.4+9.0/92.5+0.8 | 97.0/87.0 | 98.9+1.9/87.5+0.5 | 97.3/91.4 | 98.0+0.7/91.8+0.4 | 93.8/87.3 | 98.8+5.0/95.2+7.9 | 92.6/88.9 | 96.4+3.8/90.8+1.9 | 51.6/70.0 | 98.4/92.9 |
| | VisA | 90.1/82.7 | 95.2+5.1/87.6+4.9 | 87.7/78.6 | 94.5+6.8/85.2+6.6 | 93.6/88.5 | 94.9+1.3/89.7+1.2 | 83.0/76.6 | 93.4+10.4/88.9+12.3 | 86.6/85.1 | 91.3+4.7/86.7+1.6 | 50.3/67.8 | 94.8/89.8 |
| | BTAD | 90.8/73.4 | 95.4+4.6/74.9+1.5 | 91.8/64.1 | 93.9+2.1/65.7+1.6 | 93.7/74.4 | 94.8+1.1/72.9-1.5 | 94.0/76.4 | 95.5+1.5/82.6+6.2 | 83.8/71.5 | 94.8+11.0/74.2+2.7 | 54.8/15.0 | 94.2/74.8 |
| | MVTec3D | 71.4/87.0 | 84.7+13.3/92.2+5.2 | 75.1/82.6 | 83.3+8.2/87.4+4.8 | 84.9/91.8 | 85.1+0.2/92.6+0.8 | 82.9/91.2 | 86.2+3.3/89.9-1.3 | 76.4/90.5 | 81.4+5.0/92.4+1.9 | 51.9/18.9 | 84.4/93.0 |
| | MPDD | 82.9/88.8 | 94.8+11.9/94.2+5.4 | 87.7/83.4 | 92.6+4.9/92.5+9.1 | 92.2/93.5 | 90.1-2.1/94.7+1.2 | 91.6/95.7 | 94.1+2.5/98.3+2.6 | 73.0/76.2 | 91.7+18.7/94.5+18.3 | 50.3/26.4 | 94.1/95.1 |
| **ImageBind [11]** | MVTecAD | 97.9/92.6 | 98.8+0.9/92.1-0.5 | 98.5/88.9 | 98.9+0.4/88.8-0.1 | 97.8/90.7 | 98.6+0.8/91.5+0.8 | 98.7/95.6 | 99.4+0.7/96.4+0.8 | 96.0/91.2 | 98.1+2.1/91.5+0.3 | 83.5/80.3 | 98.6/92.6 |
| | VisA | 92.6/86.3 | 95.6+3.0/88.6+2.3 | 91.4/81.9 | 94.8+3.4/86.3+4.4 | 94.9/89.5 | 95.3+0.4/90.2+0.7 | 94.8/88.7 | 95.5+1.5/91.0+2.3 | 90.3/87.4 | 93.2+2.9/88.2+0.8 | 70.0/73.6 | 95.3/90.0 |
| | BTAD | 94.6/75.9 | 95.9+1.3/76.8+0.9 | 94.6/66.7 | 95.6+1.1/67.3+0.6 | 94.9/72.6 | 95.4+0.5/75.6+3.0 | 94.9/84.8 | 95.8+0.9/84.3+0.5 | 67.1/59.2 | 94.7+27.6/75.8+16.6 | 37.5/19.3 | 93.5/76.9 |
| | MVTec3D | 79.5/90.3 | 84.4+4.9/92.0+1.7 | 78.4/86.3 | 82.6+4.2/87.0+0.7 | 85.8/91.8 | 83.5-2.3/91.8+0.0 | 83.5/91.8 | 86.2+2.7/91.8+0.0 | 80.2/90.8 | 80.8+0.6/92.0+1.2 | 52.2/64.5 | 83.3/92.2 |
| | MPDD | 91.0/92.0 | 94.4+3.4/95.1+3.1 | 92.6/89.1 | 94.8+2.2/94.2+5.1 | 92.6/94.0 | 91.5-1.1/95.0+1.0 | 96.4/99.0 | 95.7-0.7/99.0+0.0 | 60.7/52.3 | 93.6+32.9/95.0+42.7 | 44.9/40.7 | 94.2/95.6 |

**Datasets.** We conduct extensive experiments on five AD datasets, including MVTecAD [2], VisA [56], BTAD [25], MVTec3D [4], and MPDD [18], to evaluate the effectiveness of our pretrained AD representations.

**Metrics.** For image-level anomaly detection, the standard metric in anomaly detection, AUROC, is used [33, 3, 2]. Nonetheless, since abnormal areas are usually smaller than normal areas, this may cause overestimated pixel-level AUROCs. This means that for small anomalies, even if they are not correctly located, the pixel-level AUROC is still high. Thus, we adopt the Per-Region-Overlap (PRO) metric proposed in [3] for anomaly localization evaluation.

**Implementation Details.** Like most pretraining works, we employ multiple backbones for anomaly representation pretraining. In this way, we can more comprehensively validate our method and provide more pretrained AD network options. Specifically, we select strong modern pretrained backbones, including CLIP [29] series, DINOv2 [26] series and ImageBind [11]. Due to different network depths, the intermediate layers to output features are different among these selected models. The details are provided in Appendix D. Then, for each layer, we construct a Feature Projector (the number of layers is 1) for learning pretrained AD representations. We fix the parameters of the backbone networks, as preserving their basic visual representation capabilities is beneficial (see Tab.2(a)). We use Adam [28] optimizer with $1e^{-4}$ learning rate to train. The batch size is set as 32 and the total training epochs are 10. The temperature hyperparameter $\tau$ and margin $\Delta r$ are set as 0.15 and 0.75. The $N_r$ is set to 2048. We use 42 as the random seed during pretraining. All training and test images are resized and cropped to $224 \times 224$.

**Setup.** Our work is to learn proprietary pretrained features for AD tasks. Thus, **we validate the effectiveness of our pretrained features by applying them into embedding-based AD methods to replace the original features**. We mainly select representative methods as the baselines, including PaDiM [8] and PatchCore [33] (they are based on feature comparison without learnable parameters, thus can intuitively reflect the quality of pretrained features). Other methods include the influential CFLOW [12] and UniAD [50], and the latest state-of-the-art method GLASS [6]. When generating residual features, we match each input image with 8 reference samples. In addition, with the norm-oriented contrastive loss, a valuable advantage of our pretrained features is that normals and anomalies can be well distinguished based on feature norm. Then, we further propose a simple AD baseline (denoted by FeatureNorm), where we directly utilize feature norms as anomaly scores.

## 4.2 Main Results

We report the dataset-level average results across their respective data subsets in Tab.1. To guarantee the rationality of result comparison, with the five backbones, we reproduce these methods based on their official open-source code and default hyperparameters and don't perform any hyperparameter tuning. By comparison, we can see that with our pretrained features, the performance of these

methods can be consistently improved on multiple datasets with various backbone networks. This demonstrates the superiority of our pretrained features and also confirms that learning proprietary pretrained features for AD tasks is effective and valuable. Especially, in some cases (*e.g.*, CLIP-Base with PatchCore, DINOv2-Base with UniAD), the original features perform poorly, while our pretrained features can bring more significant improvements. Finally, as shown in the last column (FeatureNorm[†]), another superiority of our pretrained features lies in their capacity to achieve good AD performance based on simple feature norms even without any downstream AD modeling, while the ImageNet-pretrained features don't have such an advantage.

**Explanations on Result Comparison.** Our work doesn't directly propose an AD model, but aims to provide better pretrained representations for AD tasks. Thus, applying our pretrained features in existing AD models to demonstrate performance improvement is the most reasonable way to validate the effectiveness of our work. Therefore, unlike previous AD papers that compare their proposed models with other models, we focus on the performance improvement of the same model with and without our pretrained features.

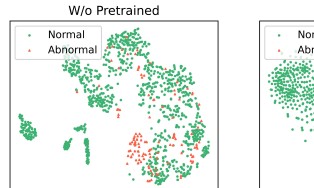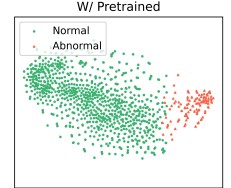

Figure 3: Feature t-SNE visualization. "w/o" and "w/" refer to without and with our pretrained features. These features are from the "capsules" class of the VisA dataset. We show more visualization results in Fig.5 in Appendix F.

**Qualitative Results.** Due to page limitation, the qualitative results are shown in Fig.4 in Appendix F.

**Intuitive Visualization.** To intuitively illustrate the effectiveness of our method, we show the t-SNE visualization of original features and our pretrained features in Fig.3. It can be found that with our anomaly representation pretraining, normal features are more compact, normal and abnormal features are more separated from each other. This intuitively demonstrates that our pretrained features are more discriminative AD representations.

## 4.3 Ablation Studies

In ablation studies, we utilize ImageBind [11] as the backbone network, followed by PaDiM and PatchCore as the baseline AD methods. Moreover, we also measure anomaly scores by feature norms (see the **Setup** part in Sec.4.1). We perform ablation experiments on the VisA dataset.

**Are pretrained features effective?** As shown in Tab.2(a), all three AD baselines can achieve better results based on our pretrained features (ExpID 2.2 and 2.7 vs. 2.1). This demonstrates that our anomaly representation pretraining method is effective.

**Are residual features better pretrained AD representations?** In Tab.2(a), the results show that residual features can bring further gains (ExpID 2.7 vs. 2.2), especially under FeatureNorm. In ResAD [46], the authors explained and verified that residual features are class-generalizable representations. Our studies also confirm the authors' statements and show that residual features are better pretrained AD representations compared to the vanilla features extracted by the backbone network.

**Contrastive Losses.** The ablation studies on contrastive losses are also in Tab.2(a). Only learning with the angle-oriented or norm-oriented contrastive loss can achieve good performance (ExpID 2.5, 2.6 vs. 2.1), while combining the two contrastive losses can further outperform each single loss consistently (ExpID 2.7 vs. 2.5, 2.6). This indicates that the two proposed contrastive losses are complementary and simultaneously optimizing the angle size and norm difference between features is more conducive to obtain good representations for anomaly detection.

**Does the backbone network need to remain fixed?** In our method, we opt to keep the backbone network fixed and only optimize the Feature Projector. Because we think that the basic visual representation capabilities possessed in the pretrained backbone are still valuable and important, and training the whole feature network on AD datasets may cause impairment to the basic visual representation capabilities. The results in Tab.2(a) verify our confirmation that the learnable backbone network will lead to performance degradation (ExpID 2.2 vs. 2.3 and ExpID 2.7 vs. 2.8). This also indicates that the AD field still needs larger-scale and better-quality datasets to support full-backbone-based anomaly representation pretraining.

Table 2: Ablation study results. (a) *Residual* represents residual features. *Non-residual* means the vanilla features extracted by the backbone network. *Angle* and *Norm* mean angle- and normal-oriented contrastive losses, respectively. *Fixed* and *Non-fixed* represent whether the backbone network remains fixed during pretraining. Note that when the backbone network is fixed, the Feature Projector is required. Otherwise, there are no learnable parameters. (b) All *Attentions* actually represent the Transformer structure (Attention + MLP).

(a) Framework ablation studies.

| ExpID | Pretrained Representations | Contrastive Losses | Feature Projector | Backbone Network | PaDiM | PatchCore | FeatureNorm |
|---|---|---|---|---|---|---|---|
| 2.1 | Non-residual | | ✓ | Fixed | 92.6/86.3 | 91.6/81.3 | 49.2/44.5 |
| 2.2 | | Angle&Norm | w/ | Fixed | 93.5/86.1 | 93.6/85.1 | 82.9/83.9 |
| 2.3 | | Angle&Norm | / | Non-fixed | 89.3/83.4 | 91.9/84.8 | 81.0/83.2 |
| 2.4 | | | ✓ | Fixed | 93.9/85.6 | 92.9/86.5 | 91.3/86.8 |
| 2.5 | | Angle | w/ | Fixed | 93.9/85.2 | 93.2/83.8 | 83.9/83.0 |
| 2.6 | Residual | Norm | w/ | Fixed | 93.7/85.3 | 90.9/84.5 | 92.4/85.1 |
| 2.7 | | Angle&Norm | w/ | Fixed | 95.4/88.7 | 94.6/87.0 | 94.2/89.0 |
| 2.8 | | Angle&Norm | / | Non-fixed | 81.3/56.0 | 78.4/32.1 | 51.9/20.7 |

(b) Architecture ablation studies for the Feature Projector.

| Architecture | PaDiM | PatchCore | FeatureNorm |
|---|---|---|---|
| Linear Projector | 93.8/87.4 | 94.0/86.8 | 87.9/82.1 |
| MLP Projector | 93.4/88.1 | 94.2/79.1 | 94.2/90.2 |
| Self Attention | 93.7/88.8 | 92.9/84.3 | 92.9/84.9 |
| Cross Attention | 94.8/88.9 | 94.6/82.9 | 93.1/86.5 |
| Self + Cross Attention | 94.8/88.7 | 94.0/80.9 | 92.9/84.4 |
| Learnable Key/Value Attention (ours) | 95.5/88.8 | 94.7/87.6 | 94.5/89.3 |

**Feature Projector Architecture.** The results are shown in Tab.2(b). Our Learnable Key/Value Attention (LKV-Attn) can outperform other network architectures. In Cross Attention, we convert normal reference features into residual features and then utilize them as Key and Value. Compared to Self Attention, Cross Attention can perform better. This indicates that only embodying normal patterns in Key and Value should be effective. In our LKV-Attn, we further adopt learnable Key and Value. Compared to limited normal patterns in Cross Attention, the learnable reference (Sec.3.3) in our LKV-Attn can adaptively learn to represent normal patterns and better cover key normal feature patterns in the residual feature distribution during pretraining.

For hyperparameter ablation studies, please see Appendix E.

## 4.4 Further Analysis

**Sample Efficiency.** We further conduct experiments about sample efficiency, where we utilize only 10% normal samples from each downstream AD dataset for training. The results are in Tab.3(a). Compared to the results in Tab.1, with less training data, our pretrained features can bring more significant performance improvement, indicating that our pretrained features are beneficial for improving the sample efficiency of downstream AD models.

Table 3: Sample efficiency and robustness analysis. † means our pretrained features are utilized in these AD models.

(a) Sample efficiency experiment results.

| Datasets | PaDiM | PaDiM† | PatchCore | PatchCore† |
|---|---|---|---|---|
| MVTecAD | 81.4/88.6 | 96.8+15.4/90.3+1.7 | 96.5/86.2 | 98.2+1.7/87.7+1.5 |
| VisA | 82.8/79.3 | 93.0+10.2/83.3+4.0 | 88.9/78.7 | 93.9+5.0/85.7+7.0 |
| BTAD | 89.2/75.2 | 94.8+5.6/76.3+1.1 | 93.1/65.5 | 94.0+0.9/68.9+3.4 |
| MVTec3D | 66.3/85.8 | 82.1+15.8/91.9+6.1 | 74.3/84.7 | 80.8+6.5/87.1+2.4 |
| MPDD | 63.1/75.6 | 91.4+28.3/94.8+19.2 | 79.1/84.4 | 90.8+11.7/93.3+8.9 |

(b) Robustness experiment results.

| Datasets | PaDiM | PaDiM† | PatchCore | PatchCore† |
|---|---|---|---|---|
| MVTecAD | 86.6/82.7 | 88.2+1.6/84.6+1.9 | 87.7/80.6 | 89.2+0.8/81.4+1.5 |
| VisA | 82.8/75.6 | 87.4+4.6/79.9+4.3 | 83.4/71.5 | 86.9+3.5/78.7+7.2 |
| BTAD | 85.4/70.9 | 90.5+5.1/73.4+2.5 | 85.1/62.4 | 86.3+1.2/65.6+3.2 |
| MVTec3D | 72.0/83.1 | 76.2+4.2/84.9+1.8 | 71.5/78.6 | 73.8+2.3/80.5+1.9 |
| MPDD | 79.8/83.8 | 84.0+4.2/86.4+2.6 | 83.4/79.1 | 84.5+1.1/85.8+6.7 |

**Robustness to Noise.** We further conduct experiments with noisy data to investigate the robustness. Specifically, we add abnormal data from the test set to the training set with a noise ratio of 0.1. We adopt the "overlap" setting from SoftPatch [19]. The results are in Tab.3(b). It can be found that the performance of AD models significantly decreases when there is noisy data. However, our pretrained features still bring performance improvement, and the magnitude is even greater compared to the results in Tab.1. This indicates that our pretrained features are more robust to noisy data than the original features. For the robustness, we think that a possible explanation may be: Although some noises are added to the training set, it is still dominated by normal features. For PatchCore as an example, the training process is to subsample a coreset. With better representation, abnormal features are more likely to be sparse outliers, making them more likely not to be sampled into the coreset. Thus, the results show better robustness.

**Reference Set Sensitivity.** Since residual features depend on few-shot normal reference samples, different normal reference samples may result in performance variations. Thus, we further investigate the sensitivity of our pretrained features to the reference set.

Table 4: Reference set sensitivity experiments.

| Datasets | PaDiM | PaDiM† | PatchCore | PatchCore† |
|---|---|---|---|---|
| MVTecAD | 97.9/92.6 | 98.6±0.14/92.1±0.05 | 98.5/88.9 | 98.8±0.09/88.4±0.37 |
| VisA | 92.6/86.3 | 95.4±0.21/88.7±0.09 | 91.4/81.9 | 94.4±0.24/86.8±0.37 |
| BTAD | 94.6/75.9 | 95.9±0.29/76.6±0.17 | 94.6/66.7 | 95.6±0.05/68.0±1.06 |
| MVTec3D | 79.5/90.3 | 84.1±0.31/92.0±0.05 | 78.4/86.3 | 82.1±0.37/87.3±0.22 |
| MPDD | 91.0/92.0 | 93.8±0.46/95.0±0.14 | 92.6/89.1 | 93.8±1.03/93.7±0.34 |

We randomly sample three reference sets (with 8 shot), conduct experiments respectively. We report variance bars in Tab.4. Overall, our pretrained features are not very sensitive to the reference set, and the performance is not significantly affected by reference variations. However, we further point out that if the reference set is not representative enough (lacking some normal patterns), the results would be more affected. We provide more discussions in Appendix C and introduce a feasible method for tackling this issue.

**Comparison to Fine-tuning Baselines.** In Appendix B.1, we further discuss and compare with previous per-dataset fine-tuning methods, such as FYD [54], MSC [31], and PANDA [30].

**Pretraining-test Data Leakage.** A non-negligible concern about our work is whether there may be anomaly data leakage between the pretraining dataset and the test datasets. We discuss this issue in Appendix B.2.

## 4.5 Few Shot Anomaly Detection

Table 5: Performance comparison with FSAD methods on the MVTecAD and VisA datasets. "∗" indicates results of these methods are from Win-CLIP [17]. "#" indicates results of these methods are from KAG-Prompt [38]. "‡" indicates these methods employ ImageBind [11] as the feature extractor.

| Setup | Method | Venue | MVTecAD | | | VisA | | |
|---|---|---|---|---|---|---|---|---|
| | | | I-AUROC | P-AUROC | PRO | I-AUROC | P-AUROC | PRO |
| 2-shot | SPADE* | arXiv2020 | 82.9 | 92.0 | 85.7 | 80.7 | 96.2 | 85.7 |
| | PatchCore* | CVPR2022 | 86.3 | 93.3 | 82.3 | 81.6 | 96.1 | 82.6 |
| | WinCLIP* | CVPR2023 | 94.4 | 96.0 | 88.4 | 84.6 | 96.8 | 86.2 |
| | AnomalyGPT#,‡ | AAAI2024 | 95.5 | 95.6 | 90.0 | 88.6 | 96.4 | 83.4 |
| | PromptAD# | CVPR2024 | 95.7 | 96.2 | 88.5 | 88.3 | 97.1 | 85.8 |
| | InCTRL | CVPR2024 | 94.0 | / | / | 85.8 | / | / |
| | ResAD‡ | NeurIPS2024 | 94.4 | 95.6 | / | 84.5 | 95.1 | / |
| | KAG-Prompt#,‡ | AAAI2025 | 96.6 | 96.5 | 91.1 | 92.7 | 97.4 | 86.7 |
| | FeatureNorm‡ (ours) | - | 95.3 | 95.6 | 90.9 | 92.4 | 97.6 | 87.5 |
| 4-shot | SPADE* | arXiv2020 | 84.8 | 92.7 | 87.0 | 81.7 | 96.6 | 87.3 |
| | PatchCore* | CVPR2022 | 88.8 | 94.3 | 84.3 | 85.3 | 96.8 | 84.9 |
| | WinCLIP* | CVPR2023 | 95.2 | 96.2 | 89.0 | 87.3 | 97.2 | 87.6 |
| | AnomalyGPT#,‡ | AAAI2024 | 96.3 | 96.2 | 90.7 | 90.6 | 96.7 | 84.6 |
| | PromptAD# | CVPR2024 | 96.6 | 96.5 | 90.5 | 89.1 | 97.4 | 86.2 |
| | InCTRL | CVPR2024 | 94.5 | / | / | 87.7 | / | / |
| | ResAD‡ | NeurIPS2024 | 94.2 | 96.9 | / | 90.8 | 97.5 | / |
| | KAG-Prompt#,‡ | AAAI2025 | 97.1 | 96.7 | 91.4 | 91.4 | 93.3 | 97.7 | 87.6 |
| | FeatureNorm‡ (ours) | - | 96.2 | 95.9 | 91.3 | 94.5 | 98.1 | 89.3 |

As stated in the **Setup** part in Sec.4.1 and shown in Tab.1, one valuable advantage of our pretrained features is that the feature norms can be directly used as anomaly scores. This means that when only few-shot normal samples are accessible, we can easily construct a few-shot AD (FSAD) method (denoted by FeatureNorm). Specifically, for an input sample, we extract pretrained residual features. Then, for a pretrained feature $x_i$, the L2 norm $||x_i||_2$ is used as the anomaly score. To demonstrate the potential of our pretrained features for FSAD, we compare our FeatureNorm with mainstream FSAD methods. We follow the 2-shot and 4-shot settings in KAG-Prompt [38], the results are in Tab.5. Compared with these competing methods, our simple FeatureNorm is comparable and even superior (on VisA). This demonstrates that the essence of FSAD still lies in the representation ability of features. With better AD representation features, we can achieve good FSAD results without the need to design elaborate methods (*e.g.*, the Kernel-Aware Hierarchical Graph in KAG-Prompt). Compared to the most relevant work, ResAD, our simple FeatureNorm can outperform it, especially on the VisA dataset. Moreover, a major advantage of our work is that we can provide better representation features for downstream AD methods, while ResAD cannot empower other AD models.

## 5 Conclusion

In this paper, we explore the problem of anomaly presentation pretraining, which is important for the anomaly detection field but has been inadvertently overlooked. We propose a novel AD representation learning framework, which learns pretrained AD representations based on residual features and consists of angle- and norm-oriented contrastive losses to fully optimize feature discrepancies. Experiments comprehensively demonstrate that our proprietary pretrained features consistently surpass the original ImageNet-pretrained features. As we all know, pretraining on ImageNet has prompted the prosperity of computer vision. Analogously, in the AD field, a valuable question is what kind of representations can serve as the fundamental (general) representation for anomaly detection. Furthermore, more attention could be paid to anomaly representation pretraining in future work, rather than constantly focusing on designing more sophisticated AD models dependent on ImageNet-pretrained networks. The limitations of our method are discussed in Appendix B.5.

## Acknowledgments

This work was supported in part by the National Natural Science Fund of China (62371295) and the Science and Technology Commission of Shanghai Municipality (22DZ2229005).

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

# Appendix

## A    Significance of Anomaly Representation Pretraining

Strictly speaking, from the implementation perspective, our method is not true "pretraining" as it didn't achieve pretraining the whole backbone network. However, the usage of "pretraining" has two reasons: 1) We are indeed following the basic paradigm of pretraining: learning on a large-scale dataset, and the learned features can be transferred to downstream datasets and models. 2) More importantly, we want to emphasize that in the current development stage of anomaly detection, fundamental representations specifically learned for anomaly detection are of great significance. Current mainstream AD models mostly use pretrained networks to extract features. However, regardless of supervised or self-supervised pretraining on natural images, the pretraining process does not match the goal of anomaly detection. From the perspective of the development prospects of AD, it's necessary for AD tasks to have specialized pretrained features instead of continuously using basic pretrained features. Our work is only an early exploration, and we hope to inspire more future works to focus on anomaly representation pretraining.

## B    More Discussions

### B.1    Discussions with Fine-tuning Baselines

We further provide a discussion with the previous finetune-based baseline methods, such as FYD [54], MSC [31], and PANDA [30]. These methods aim to finetune pretrained features to adapt to the target distribution in specific AD datasets. As the FYD paper does not provide open-source code, we can only compare with the results provided in the paper. In FYD, anomaly detection is achieved based on PaDiM. The paper provides the results on MVTecAD, which are 97.7/98.2 (image-level/pixel-level AUROCs). Based on PaDiM, the results of our method are 98.1/98.5. In PANDA and mean-shift contrastive loss (MSC), feature adaptation is based on global features, so only image-level anomaly detection is performed. The two papers provide image-level AUROC on MVTecAD, with values of 86.5 and 87.2. For a more reasonable comparison, we utilized the open-source code of MSC and trained an adapted network (based on WideResNet50) on MVTecAD. Then, we utilized PaDiM for AD modeling, and the obtained results are 96.8/97.0. Thus, compared with PANDA, MSC, and FYD, our method has advantages in providing better features for AD models. Moreover, adaptive feature fine-tuning requires specialized fine-tuning on each dataset. By comparison, a more important advantage of anomaly representation pretraining is that pretrained features can be used on multiple downstream AD datasets without the need for fine-tuning on each dataset.

It is crucial to point out that FYD does not utilize large-scale AD datasets for pretraining when improving feature representations. Thus, what impact does the dataset scale have on feature fine-tuning? To answer this, we further perform pretraining on the MVTecAD dataset. The results are in Tab.6.

Table 6: Experimental analysis of the impact of pretraining dataset scale.

| Datasets | PaDiM (RealIAD) | PaDiM ( MVTecAD) | PatchCore (RealIAD) | PatchCore (MVTecAD) |
|---|---|---|---|---|
| MVTecAD | 98.9/92.1 | 97.7/92.1 | 98.9/88.8 | 98.8/88.0 |
| VisA | 95.6/88.6 | 93.0/85.6 | 94.8/86.3 | 93.4/86.0 |
| BTAD | 95.9/76.8 | 95.6/76.6 | 95.6/67.3 | 94.7/66.5 |
| MVTec3D | 84.4/92.0 | 82.8/91.2 | 82.6/87.0 | 80.6/87.3 |
| MPDD | 94.4/95.1 | 92.9/93.5 | 94.8/94.2 | 93.1/93.4 |

**Impact of Pretraining Dataset Scale.** It can be seen that with pretraining on MVTecAD, the results are overall lower than the results obtained based on RealIAD pretraining. Thus, we think that the scale of the dataset will have an impact, a larger scale is more advantageous for pretraining (better transferability of the learned features). This also reflects that the AD field still needs larger and higher-quality datasets in the future to better support anomaly representation pretraining.

However, in FYD, the transferability of the fine-tuned features is not discussed. In FYD, after fine-tuning on MVTecAD, the fine-tuned model is effective on MVTecAD, but it has not been verified whether it is still effective on other datasets. We think that FYD requires specialized fine-tuning on each dataset. The reason is that the fine-tuning process in FYD is mainly to accomplish spatial

alignment (beneficial to the PaDiM method) on feature maps. As the spatial distributions of objects in different datasets are different, the learned affine transformation parameters in one dataset should be hard to apply to other datasets. Therefore, even if we employ FYD to fine-tune on RealIAD, the adapted network cannot be directly applied to other downstream datasets. Nevertheless, in our paper, the results in Tab.1 have comprehensively verified that the pretrained features have good transferability in downstream AD datasets.

## B.2 Discussions on Data Leakage

A concern about our work is whether there may be anomaly data leakage between the pretraining dataset and the test datasets. We think that the data leakage issue does not need to be concerned. Because the product categories in RealIAD are completely different from those in the test AD datasets, and the anomaly types are also quite different. Even if there are some similar types of anomalies (*e.g.*, broken), the visual appearance displayed on different products will also be different. For data leakage, it's hard to calculate a numerical indicator. Due to different products, the similarities between pretraining images and test images are all low, making it hard to distinguish what constitutes anomaly data leakage. In Tab.6, we report results of both pretraining and testing on the MVTecAD dataset. This is a serious data leakage, but compared to RealIAD pretraining, it doesn't bring significant performance improvement. This indicates that data leakage is not a critical issue for the performance gains in our work.

## B.3 Discussions with SPD

We further provide a discussion with a previous pretraining-related method, SPD [56]. In the paper spot-the-difference (SPD), the authors propose a data augmentation strategy called SmoothBlend to create slight local differences (or called pseudo-anomalies). Then, the authors create pseudo-anomalies on natural images from ImageNet to use as negative samples in contrastive learning. The model is pretrained on ImageNet, while our model is pretrained on a real AD dataset, RealIAD. (1) Compared to pseudo-anomalies, the negative samples used in our method are more realistic and more in line with the abnormal characteristics in real-world scenarios. (2) In SPD, similar to classical SSL methods (SimCLR, MoCo), contrastive loss is constructed based on global features, while we construct contrastive loss based on more fine-grained local features. Although SPD mentioned that it can promote local sensitivity compared to SimCLR, MoCo, etc., it still may lead to the loss of information about local pseudo-anomalies in the global features, as the local augmentation is slight. For fine-grained task like anomaly detection, contrastive learning based on fine-grained local features should be more suitable. (3) SPD was evaluated only on the ResNet-50, PaDiM, and VisA and MVTecAD datasets, while we evaluate our method on more backbones, AD models, and AD datasets.

## B.4 More Discussions with ResAD

We think that we should further discuss some similarities and differences between our work and ResAD [46]. Compared to ResAD, we mainly employ the wonderful residual features proposed in ResAD as our pretrained AD representations. Because we expect that pretrained features can serve as general features in anomaly detection, namely, it's best for pretrained representations to be domain-invariant. As the authors explained in [46], the residual features can be regarded as class-generalizable representations, which satisfy our expectations just right. The results in Tab.2(a) also demonstrate that residual features are better pretrained AD representations compared to the vanilla features extracted by the backbone network. Our studies also further confirm the authors' opinions in [46] that residual features have the potential to be general representations in anomaly detection. However, our work and ResAD are clearly two different works, with differences in motivations, concerned problems, and implementation methods.

(1) **Different Motivations and Concerned Problems**. ResAD explores the problem of class-generalizable anomaly detection. The authors mainly aim to learn a generalizable AD model that can be directly applied to new classes to accomplish anomaly detection. By comparison, our work focuses on a very valuable but inadvertently overlooked problem: anomaly representation pretraining. We aim to learn better and proprietary representation features for AD tasks through contrastive learning pretraining on large-scale AD datasets.

(2) **Class-generalizable Anomaly Detection vs. Anomaly Representation Pretraining.** Although class-generalization anomaly detection can also be seen as pretraining an AD model on a dataset and then applying it to new datasets. However, it's still significantly different from anomaly representation pretraining, as anomaly representation pretraining aims to provide better AD representation features for downstream AD methods, while class-generalizable anomaly detection (ResAD, InCTRL [55]) cannot empower other AD models. This is why in the experiments, we didn't perform reverse setting (train on the combination of MVTecAD, VisA and other datasets and eval on RealIAD). We need to validate the effectiveness of our pretrained features in existing AD methods, rather than verifying the cross-class generalization ability of AD models. The models in Tab.1 are also first trained on normal samples with our pretrained features, and then evaluated.

(3) **Different Implementation Methods**. Our work doesn't construct a model that can be directly used for anomaly detection, while providing pretrained representation features for AD tasks. Our pretraining framework is based on contrastive learning. To fully optimize discrepancies between normal and abnormal features, we propose angle- and norm-oriented contrastive losses from the feature similarity perspective. ResAD constructs a class-generalizable AD model that accomplishes anomaly detection based on normalizing flow model [21] to learn residual feature distribution. In addition, ResAD uses fixed normal samples as reference to generate residual features, without considering the semantic misalignment issue. We realize this issue and provide an improvement (the semantic-aligned reference matching module, Appendix C) for generating semantic-aligned residual features.

(4) As for performance, we compare the FSAD performance with ResAD. As shown in Tab.5, our simple FeatureNorm is better than ResAD, especially on the VisA dataset. Moreover, compared to the more sophisticated normalizing flow modeling in ResAD, calculating feature norms in our method is quite simple. This also demonstrates the value of our pretrained features.

## B.5    Limitations

In this work, unlike conventional AD research works, we explore a very valuable but inadvertently overlooked problem: anomaly representation pretraining. We propose a novel AD representation learning framework to learn proprietary pretrained features for AD tasks. Even if incorporating our pretrained features into embedding-based AD methods can manifest good performance improvement on five AD datasets and five backbones (Tab.1), there are still some limitations of our work.

One limitation of our work is that we fix the backbone network and only optimize the Feature Projector during training. However, we think that for an ideal feature extraction network in AD tasks, the backbone part should be specially designed based on the characteristics of anomaly detection and effectively learned during pretraining. For example, when encoding one image, the backbone network can simultaneously consider multiple normal reference samples to increase the normal context patterns. Then, the contrast between normal and abnormal can be implicitly embodied in the network encoding process. Thus, the encoded normal and abnormal features would be more discriminative.

Another limitation is that our pretrained features can only be incorporated into embedding-based AD methods. Other AD methods that are not based on pretrained feature extractors (*e.g.*, diffusion-based) cannot benefit from our method. Therefore, future work should also focus on exploring pretraining frameworks for more AD methods.

## B.6    Computational Cost

In this work, we accomplish anomaly detection based on some embedding-based AD methods. The computational cost of a complete AD model is determined by both the feature extractor and the subsequent AD method. Compared to using ImageNet-pretrained features, our method mainly introduces extra cost in the Feature Projector. For different backbones, the computational costs of the Feature Projector are different. With the image size fixed as $224 \times 224$, we calculate the number of parameters and computation FLOPs in the Feature Projector. The results are shown in Tab.7.

Table 7: Computational cost of the Feature Projector with different backbone networks.

| | DINOv2-Base | DINOv2-Large | CLIP-Base | CLIP-Large | ImageBind |
|---|---|---|---|---|---|
| parameters | 21.3M | 37.8M | 21.3M | 37.8M | 59M |
| FLOPs | 137.7G | 309.2G | 137.7G | 309.2G | 483.2G |

## B.7 Social Impacts and Ethics

As a work for learning robust and discriminative pretrained representations for anomaly detection, the proposed method does not suffer from particular ethical concerns or negative social impacts. All datasets used are public. All qualitative visualizations are based on industrial product images, which don't infringe personal privacy.

## C Semantic-Aligned Reference Matching

**Discussions on Reference-Set Sensitivity.** Residual features are mainly relied on mutually eliminating class-related normal patterns in features through subtraction (please see the ResAD paper [46] for more explanations), for being class-generalizable. However, if normal images have significant variability, it may cause the few-shot reference samples to be not representative enough (lacking some class-related normal patterns). This may lead to certain class-related normal patterns in the input feature can't be effectively eliminated. Then, the discriminability of residual features under few-shot settings will be compromised.

For practical applications, this issue should be particularly focused and reasonably addressed. Of course, the simplest solution is to increase the number of reference samples. This is feasible, as in practical applications, the number of reference samples is usually not as strict as the 8-shot. However, in practical applications, we expect that the reference samples can fully represent their class, so it's best to have sufficient differences between the reference samples. Thus, the sample selection strategy cannot be random. A feasible method is to first cluster all available normal samples into different clusters based on a clustering algorithm (e.g., KMeans). Then, based on the number of reference samples, we evenly distribute it to each cluster. When selecting from a cluster, we can prioritize selecting samples closer to the center. Another feasible strategy is to match spatially aligned samples as reference samples for each input sample. Below, we provide a corresponding approach.

**Residual Features with Semantic-Aligned Reference Matching.** When constructing residual features, it's required to first match the nearest feature from the reference feature bank. To maintain the efficiency of residual feature generation, the reference feature bank cannot be too large. Thus, in [46], the authors use fixed few-shot normal samples to construct the reference feature bank. Although efficiency is guaranteed, it may sacrifice the representation ability of residual features. In real-world scenarios, different normal samples of the same class may have semantic differences (*e.g.*, variable object components or shapes, misaligned objects). When the semantic differences among normal samples are too large, it can cause fixed few-shot normal samples can't cover all normal patterns, thereby the reference feature bank may not be representative enough. To address this, we can select proper semantic-aligned (*i.e.*, the components, shape, and orientation of the object in two images are relatively consistent) reference samples for each input image. In specific, we propose a lightweight semantic-aligned reference matching method, which calculates statistical histograms (distances from local features to clustering centers) for the image and then measures the global distance between two images by the KL divergence between histograms. Two aligned images usually have close spatial feature distribution, so the histograms are similar, namely, smaller KL divergence. As histograms are low-dimensional, the calculation of KL divergences has a very low cost. Thus, the matching module will not introduce too much cost. Finally, we indicate that this module is not used during pretraining, while it can be used when we apply pretrained features to downstream AD datasets.

**Semantic-Aligned Reference Matching.** We describe our semantic-aligned reference matching method that can retrieve semantic-aligned samples from the normal sample set for the input image $I$. Given total $N$ normal images, we extract first layer features with a pretrained lightweight network (*e.g.*, ResNet-18), the corresponding features are denoted as $\{F_i \in \mathbb{R}^{H_1 \times W_1 \times C_1} | i = 1, 2, \ldots, N\}$. Then, for each image, the feature map $F_i$ is evenly divided into $S \times S$ grids as:

$$F_i^{u,v} \in \mathbb{R}^{\frac{H_1}{S} \times \frac{W_1}{S} \times C_1}, u, v = 1, 2, \ldots, S \tag{8}$$

For each grid, we collect all raw patch features of different normal images together into the feature set $\mathcal{P}^{u,v} = \{\text{Flatten}(F_i^{u,v})^k \in \mathbb{R}^{C_1} | k = 1, 2, \ldots, NH_1W_1/S^2\}$. The K-means clustering algorithm is performed on $\mathcal{P}^{u,v}$ to obtain $N_c$ clustering centers $\mathcal{C}^{u,v} = \{c_k \in \mathbb{R}^{C_1} | k = 1, 2, \ldots, N_c\}$. For each grid, we can obtain $N_c$ clustering centers, and then the total number of centers is $S^2 N_c$.

The feature map $F_i$ and clustering centers $\{\mathcal{C}^{u,v} \in \mathbb{R}^{N_c \times C_1} | u, v = 1, 2, \ldots, S\}$ are further utilized to calculate the block-wise statistics for each normal image $I_i$. Specifically, in the $(u, v)$-th grid, we calculate the cosine similarity between the grid feature map $F_i^{u,v} \in \mathbb{R}^{\frac{H_1}{S} \times \frac{W_1}{S} \times C_1}$ and all centers $\mathcal{C} \in \mathbb{R}^{S^2 N_c \times C_1}$. Then we obtain $H_1W_1/S^2$ histograms denoted by $H_i^{u,v} = \{h_k \in \mathbb{R}^{S^2 N_c} | k = 1, 2, \ldots, H_1W_1/S^2\}$, where $h_k$ denotes the cosine distance histogram of $k$-th feature in the grid with respect to the codebook $\mathcal{C}$. The histograms are then normalized so that $||h_k||_1 = 1, \forall k$. Then, for the input image $I$ and the $i$-th normal image $I_i$, we can calculate the KL divergence between their histograms to measure the spatial alignment degree between them. Specifically, the calculation of KL divergence is based on grid, we calculate the block-wise KL divergence $D_i^{u,v} = KL(H_i^{u,v}, H^{u,v})$ for each block $(u, v)$, where $H^{u,v}$ means the histograms in $(u, v)$-th grid of the input image. As in one grid, there are $H_1W_1/S^2$ histograms, the shape of $D_i^{u,v}$ should be $\mathbb{R}^{H_1W_1/S^2}$. If the two grids are semantically similar, the maximum KL divergence needs to be small enough. Therefore, we only select the maximum KL divergence $d_i^{u,v}$ from $D_i^{u,v}$ as semantic alignment degree between two grids, $d_i^{u,v} = \max(D_i^{u,v})$. The global semantic alignment degree between $I$ and $I_i$ is estimated as:

$$D_i^{align} \triangleq \frac{1}{S^2} \sum_{u=1}^{S} \sum_{v=1}^{S} d_i^{u,v} \tag{9}$$

Based on the global semantic alignment degree, we can sort the global distances between the input image and $N$ normal images and then retrieve the top-K neighbor normal images (we denote the set of indices as $\iota$) as few-shot normal reference images $\mathcal{N}_{ref} = \{I_{\iota_1}, I_{\iota_1}, \ldots, I_{\iota_K}\}$. In implementation, the hyperparamters $S$, $N_c$, and $K$ are set as 5, 5, and 8, respectively.

**Further Discussion.** Employing the semantic-aligned reference matching approach is to provide proper semantic-aligned reference samples for each input image. Generally speaking, such reference samples should be more reference-informative, especially when the test images are highly non-aligned. Moreover, in practical usage, when inputting a test image, the semantic-aligned reference matching approach will not introduce too much extra cost for the inference process. We can employ it as a preprocessing step by calculating histograms for all collected normal samples and storing these histograms. Then, the input image only needs to calculate its own histograms and calculate the block-wise KL divergences with the stored histograms, and the calculation of KL divergences has a very low cost. When using ResNet-18 as the lightweight network, the cost of calculating the global semantic alignment degree between $I$ and $I_i$ is only about 0.1 GFLOPs.

## D  Implementation Details

Like most pretraining works, we employ multiple backbones with different architectures and parameter scales for anomaly representation pretraining. Specifically, we select DINOv2-B [26], DINOv2-L [26], CLIP-B [29], CLIP-L [29], and ImageBind [11] as the backbone network. Due to different network depths, the intermediate layers to output features are different among these models. DINOv2-B and CLIP-B have a total of 12 layers, and we use the [3, 6, 9, 12] layers to output features. Network layers in DINOv2-L and CLIP-L are 24, and features from the [6, 12, 18, 24] layers are used for training. For ImageBind, the outputs from the [8, 16, 24, 32] layers are used for training.

**Implementation Details About the Results in Tab.1.** In Tab.1, to guarantee the rationality of result comparison, with the five backbones, we reproduce these methods based on their official open-source code. For each method, we use the default hyperparameters and don't perform any hyperparameter tuning. When applying our pretrained features in these methods, we make sure that we only replace the original features without any additional modifications.

## E  Hyperparameter Sensitivity

In this section, we conduct more ablation experiments on the main hyperparameters in our method to illustrate their sensitivity. Following the main text, in hyperparameter ablation studies, we also utilize

ImageBind as the backbone network and produce results with PaDiM, PatchCore, and FeatureNorm as the baseline AD methods.

**Temperature Hyperparamter** $\tau$**.** In Tab.8, we ablate the temperature hyperparameter $\tau$ in the angle-oriented contrastive loss (see Eq.(2)). The results show that larger $\tau$ (*i.e.*, $> 0.3$) will lead to degraded results. When $\tau$ is set to 0.3, the results produced by FeatureNorm are the best, but the PRO result under PatchCore is poor. When $\tau$ is set to 0.1, the results produced by Patchcore are the best, but the results under FeatureNorm are poor. Therefore, by overall consideration, setting the temperature hyperparameter $\tau$ to 0.15 is the most suitable choice.

Table 8: Ablation studies about the temperature hyperparameter $\tau$ in the angle-oriented contrastive loss (see Eq.(2)). $\cdot/\cdot$ means image-level AUROC and PRO.

| Method \ $\tau$ | 0.1 | 0.15 | 0.2 | 0.25 | 0.3 | 0.5 |
|---|---|---|---|---|---|---|
| PaDiM | 95.4/87.8 | 95.5/88.2 | 95.4/87.9 | 95.3/87.8 | 95.1/87.5 | 94.9/87.3 |
| PatchCore | 94.4/87.5 | 94.6/87.0 | 94.3/87.4 | 94.5/86.5 | 94.5/86.5 | 94.6/85.5 |
| FeatureNorm | 93.3/87.7 | 93.9/88.1 | 93.9/87.9 | 94.1/87.9 | 94.1/88.1 | 93.8/87.5 |

**Margin Hyperparamter** $\Delta r$**.** In Tab.9, we ablate the margin hyperparameter $\Delta r$ in the norm-oriented contrastive loss (see Eq.(5)). The results show that our method is not very sensitive to the margin $\Delta r$. We think the reason should be that in the norm-oriented contrastive loss, we only need to set a certain margin to ensure that the norm of normal and abnormal features can be distinguished, rather than having to set a large enough margin. When $\Delta r$ is set to 0.75, the results under three baseline methods are overall better compared to other $\Delta r$ values. Therefore, we decide to set the margin hyperparameter $\Delta r$ to 0.75 in our method.

Table 9: Ablation studies about the margin hyperparameter $\Delta r$ in the norm-oriented contrastive loss (see Eq.(5)), $r' = r + \Delta r$).

| Method \ $\Delta r$ | 0.1 | 0.2 | 0.3 | 0.5 | 0.75 | 1.0 |
|---|---|---|---|---|---|---|
| PaDiM | 95.5/88.2 | 95.5/88.3 | 95.5/88.1 | 95.5/88.1 | 95.5/88.1 | 95.5/88.0 |
| PatchCore | 94.6/87.0 | 94.6/87.2 | 94.6/86.6 | 94.6/87.4 | 94.6/87.5 | 94.3/87.2 |
| FeatureNorm | 93.9/88.1 | 93.9/88.0 | 93.9/88.0 | 93.9/88.0 | 94.0/88.2 | 94.0/88.4 |

**Loss Weighting Coefficient** $\lambda$**.** The results are shown in Tab.10. It can be found that with $\lambda$ set to 1, the best results can be achieved.

Table 10: Ablation studies about the loss weight $\lambda$ in the total loss (see Eq.(7)).

| Method \ $\lambda$ | 0.5 | 1 | 1.5 | 2 | 3 | 5 |
|---|---|---|---|---|---|---|
| PaDiM | 95.4/88.0 | 95.5/88.1 | 95.5/88.0 | 95.4/87.7 | 95.3/87.5 | 95.2/87.2 |
| PatchCore | 94.3/87.8 | 94.6/87.5 | 94.5/87.0 | 94.6/87.3 | 94.6/87.6 | 94.5/87.2 |
| FeatureNorm | 93.5/88.0 | 94.0/88.2 | 93.9/88.1 | 93.9/87.9 | 93.9/88.1 | 94.1/88.0 |

**Number of Layers in the Feature Projector** $L$**.** The results are shown in Tab.11. The results indicate that more network layers don't bring obvious performance improvement, instead, the PRO results under PatchCore are lower than the counterpart (87.5) when $L$ is 1. Considering that more network layers require more computational cost, we only construct one attention layer in the Feature Projector.

**Number of Learnable Reference Representations** $N_r$**.** The results are shown in Tab.12. It can be found that with $N_r$ set to 2048, the PRO result under PatchCore is better than others, and the results under PaDiM and FeatureNorm are also good. Thus, the number of learnable reference representations is set to 2048 by default.

**Number of Reference Samples during Training** $K$**.** The results are shown in Tab.13. Compared to a fixed number of reference samples, combining multiple reference sample numbers and randomly selecting (Rand (1, 4, 8)) each time can achieve significant performance improvement. To vary the

Table 11: Ablation studies about the number of attention layers in the Feature Projector.

| Method \ $L$ | 1 | 2 | 3 |
|---|---|---|---|
| PaDiM | 95.5/88.1 | 95.7/88.2 | 95.7/88.4 |
| PatchCore | 94.6/87.5 | 94.9/86.4 | 95.0/86.7 |
| FeatureNorm | 94.0/88.2 | 94.0/88.1 | 93.9/88.2 |

Table 12: Ablation studies about the number of learnable reference representations in the Feature Projector.

| Method \ $N_r$ | 256 | 512 | 1024 | 1536 | 1792 | 2048 | 2560 |
|---|---|---|---|---|---|---|---|
| PaDiM | 95.4/88.1 | 95.6/88.3 | 95.5/88.1 | 95.6/88.3 | 95.6/88.1 | 95.5/88.0 | 95.5/87.9 |
| PatchCore | 94.8/87.3 | 94.8/86.7 | 94.6/87.5 | 94.5/86.5 | 94.6/87.4 | 94.7/88.0 | 94.6/87.3 |
| FeatureNorm | 94.0/88.1 | 94.2/88.3 | 94.0/88.2 | 93.8/88.0 | 93.6/87.9 | 93.8/88.1 | 93.7/88.1 |

number of reference samples during training, we think that it is beneficial for increasing residual feature diversity.

Table 13: Ablation studies about the number of reference samples used during training. Rand(1, 4, 8) means that we randomly select 1, 4, or 8 reference samples for each input sample.

| Method \ $K$ | 1 | 2 | 4 | 8 | Rand(1, 4, 8) |
|---|---|---|---|---|---|
| PaDiM | 95.5/88.0 | 95.3/88.0 | 95.3/88.1 | 95.2/88.7 | 95.5/88.8 |
| PatchCore | 94.7/88.0 | 94.4/87.5 | 94.6/87.7 | 94.3/87.5 | 94.7/87.6 |
| FeatureNorm | 93.8/88.1 | 94.3/89.0 | 94.5/89.3 | 94.2/89.2 | 94.5/89.3 |

# F  Qualitative Results

In this section, we show some qualitative results. To avoid redundant qualitative figures, we select to generate anomaly score maps based on PatchCore as it is representative enough. PatchCore is based on feature comparison without learnable parameters, and thus can intuitively reflect the quality of pretrained features. In addition, we utilize CLIP-L as the backbone network. The qualitative results are shown in Fig.4. It can be seen that by employing our pretrained features, the PatchCore method can generate better anomaly localization maps compared to using the original ImageNet-pretrained features. "PatchCore w/o pretrained" may generate many false positives in normal regions (*e.g.*, line 1, right part of lines 4 and 5). By comparison, "PatchCore w/ pretrained" can effectively avoid false positives in normal regions and locate anomalies more accurately (*e.g.*, left part of lines 3, 4, and 5).

In Fig.5, we show more feature t-SNE visualization results.

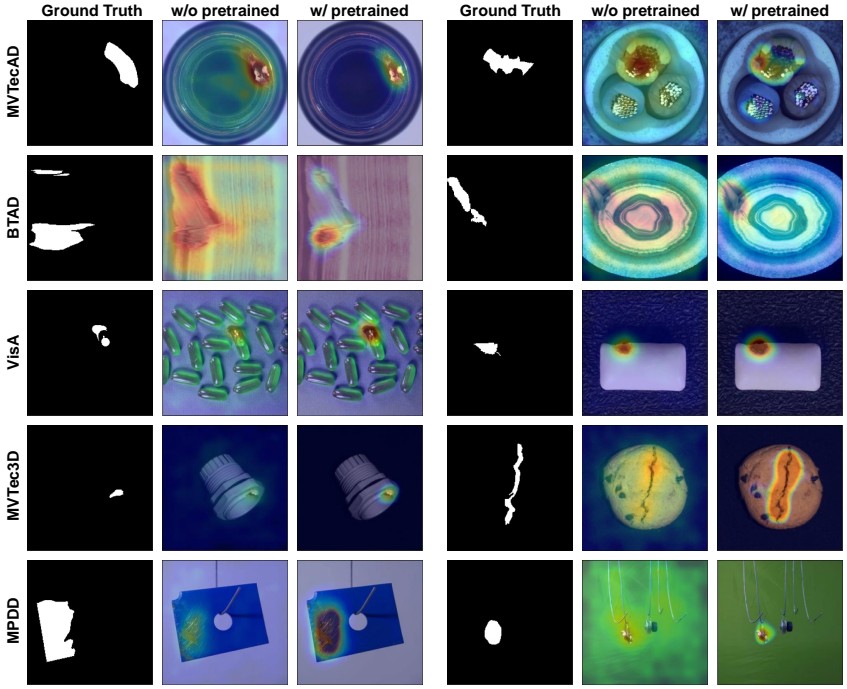

Figure 4: Qualitative results. The anomaly score maps are generated by PatchCore with CLIP-L as the backbone network. "w/o pretrained" and "w/ pretrained" refer to without and with our pretrained features.

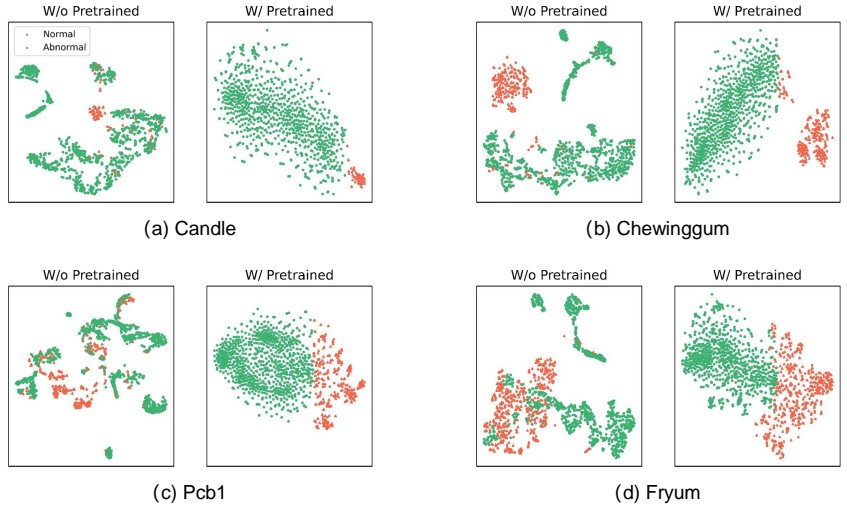

Figure 5: Feature t-SNE visualization. For (a), (b), (c), and (d), the features are from the "candle", "chewinggum", "pcb1", and "fryum" classes from the VisA dataset.

