# OpenReview forum: "ADPretrain: Advancing Industrial Anomaly Detection via Anomaly Representation Pretraining"
_NeurIPS.cc/2025/Conference — NeurIPS 2025 poster_

### Official Review · Reviewer_oLCH · 2025-06-29

**Clarity:** 1
**Significance:** 2
**Originality:** 2
**Rating:** 3
**Confidence:** 4

**Summary:**

This work proposes a feature extractor grounded in existing pretrained model (such as ResNet50, CLIP on ImageNet) for industrial anomaly detection (IAD). The proposed framework designs an angle- and norm-oriented contrastive loss to train the Feature Projector based on RealIAD, a large industrial anomaly detection dataset. The empirical results indicate that the features produced by the trained Feature Projector improve the detection performance of five embedding-based AD methods in comparison to using their original features.

**Questions:**

1. How to determine the hyperparameter radius r? Authors have mentioned that it was set to 0.4 in the experiments.

2. How large is the feature dimension? Does different dimension have a significant impact on detection performance?

3. It needs to be clarified how to construct a few-shot AD method by the feature norm.

Other questions see Weaknesses

**Ethical Concerns:**

["NO or VERY MINOR ethics concerns only"]

**Final Justification:**

I acknowledge the authors’ thoughtful response. The rebuttal has addressed some of my concerns, especially the presented results regarding $N_r$ and the conventional AD methods.  Based on this consideration, I decide to upgrade my score.

**Limitations:**

1. Building on the optimization objective and empirical results, the features produced by the proposed framework show the strong distinguishability. In such situations, it should be explored when directly applying the feature vectors to the classical AD algorithms, such as OCSVM, LOF, Isolation Forest, KNN etc. It is possible to obtain good performance due to the results of FeatureNorm. If the features indeed achieve better or comparable performance on the conventional AD algorithms than other features even the end-to-end deep methods, the proposed framework will produce much higher social impact due to the robustness and low computational cost of these conventional methods.

2. Two simple Feature Projector designs should be explored, (1) concatenating a MLPs (learnable parameters) behind the pretrained model (fixed parameters) and then fine-tuning the MLPs; (2) fine-tuning only the readout layer of the pretrained model when freezing all other parameters

**Quality:**

2

**Strengths And Weaknesses:**

**Strengths:**

1. The proposed framework effectively utilizes the RealIAD and the existing pretrained feature extractors for industrial anomaly detection.

2. The empirical results demonstrate the effectiveness of the proposed framework.


**Weaknesses:**

1. Ambiguous and misleading writing and statements

(1)	The “Unsupervised Anomaly Detection” is highlighted in Title of the manuscript. However, the proposed framework is a feature extractor, does not direct relationship with “Unsupervised learning” and the training process of the Feature Projector is based on ground truth of RealIAD (supervised learning). In fact, the proposed method is only designed to obtain the features and it doesn’t care about the specifics of the downstream AD task, whatever supervised or unsupervised.

(2)	This work only focuses on industrial anomaly detection (IAD). Thus, the research field should be clarified rather than directly using “anomaly detection” in the beginning. For instance, the description in line 1-3 is incorrect when considering other data types (time-series, graph, tabular) or application scenarios.

(3)	This work highlights the “Pretraining” in the many places of the manuscript, which makes me think a totally novel pretrained model for IAD in the beginning. However, the experimental settings show that the backbone (architecture) and even the parameters of the pretrained models are remained accurately. In other words, this work just fine-tunes the existing pretrained models on a vertical field dataset (RealIAD) for industrial anomaly detection via slotting a Feature Projector. I suggest that authors revise some statements.

2. Unclear technique details

(1)	On Angle-Oriented Contrastive Loss:
The training process of Feature Projector is supervised, that is it is easy to get positive-pairs and negative-pairs, why still using data augmentation to produce more positive-pairs? And, how to ensure the augmented samples closer to normal data than abnormal data? Especially, if the forming of anomalies is similar with the used data augmentation techniques, for example, food spoilage and random color jitters, it may cause the wrong optimization direction.

(2)	Norm-Oriented Contrastive Loss:
Why using pseudo-Huber distance? Any defects or problems if using a simple Euclidean? I understand the first term of the loss, but why adding a multiplicative term $e^{d_j}$?

(3)	Learnable Key/Value Attention and Feature Projector:
The current contents are hard to understand in Section 3.3. To improve the readability, it should describe the function of feature projector using standard formulars step by step.
I am curious about the learnable reference. After training, whether it was be fixed for extracting features for new datasets? How to understand the learnable reference?

(4)	Authors need to check and clarify the optimized object. In line140, which one is the residual representation, x or r? And all the loss functions use x as optimized object.

3. The fairness of comparison

In Appendix D Hyperparameter Sensitivity, authors explore the effects of different hyperparameter settings for detection performance. There are several concerns as follows:

(1)	What and where is the test data?

(2)	It is unreasonable that the hyperparameters selection depends on the performance on the used embedding-based AD methods. The proposed framework is a feature extractor and it should be method-agnostic, like a pretrained ResNet-50 on ImageNet rather than method-dependent, otherwise, the proposed framework is just tailored for certain AD methods.

(3)	Why does the performance look like quite robust (<1% fluctuation) in almost all cases for the changing of the hyperparameters?

(4)	For selection of $N_r$, I noticed the performances are quite close for $N_r=256$ and $N_r=2048$. It should extend the range from 1-256 to more suitable selection.

4. Statistical significance and reproducibility of the results

(1)	The empirical results do not report any average values and standard variance.

(2)	The code is not provided.

---

> ### Author Rebuttal · Authors · 2025-07-29
>
> **[To W1]** (1) The use of "unsupervised anomaly detection" mainly refers to downstream AD tasks, as unsupervised AD is the most basic paradigm in anomaly detection (AD), and our experiments are also based on unsupervised AD methods. This "unsupervised" doesn't mean that the training process is unsupervised.
>
> (2) Combined with (1), we think a good revision would be to use "industrial anomaly detection" in the title and the main text.
>
> (3) The use of "pretraining" has two reasons: 1) We are indeed following the basic paradigm of pretraining: learning on a large-scale dataset, and the learned features can be transferred to downstream datasets and models. However, fine-tuning generally refers to fine-tuning a model on a dataset, and then applying the model to the dataset. We think that pretraining should not limit that must train the whole network. 2) More importantly, we want to emphasize that in the current development stage of AD, pretrained representations specifically learned for AD are of great significance. If it's revised to "pretrained features fine-tuning", the significance of this research direction cannot be well reflected. Current mainstream AD models mostly use pretrained networks to extract features. However, regardless of supervised or self-supervised pretraining on natural images, the pretraining process does not match the goal of AD. From the perspective of the development prospects of AD, it's necessary for AD tasks to have specialized pretrained features instead of continuously using basic pretrained features. We hope to inspire more future works to focus on anomaly representation pretraining.
>
> **[To W2]** (1) **We respectfully argue that you may misunderstand our Angle-Oriented Contrastive Loss**. The data augmentation is to create a positive pair for each feature $x_i$. You may think that for a normal/abnormal feature, we use all other normal/abnormal features as positive pairs. But it's not like that (you can look the Eq.(2) again). For $x_i$, we only use $x_{i\prime}$ ($x_{i\prime}$ is the feature from the augmented image, its position in the augmented image is the same as the position of $x_i$ in the original image) as the positive pair. Other features with different labels from $x_i$ are used as negative pairs. This ensures that we only perform contrast between normal and abnormal. This is the same as in classical contrastive learning methods, where for a sample, only the augmented sample is treated as the positive pair, while other samples are treated as negative pairs.
>
> Anomalies are usually different from each other, so abnormal features cannot be mutually used as positive pairs to attract together. For normal, there are also differences among different normal patches, so normal features also should not be mutually used as positive pairs.  Thus, for $x_i$, we only use the feature $x_{i\prime}$ as the positive pair. Another reason is that if we use all features with the same labels to $x_i$ as positive pairs, this will cause numerical issues in the loss (i.e., appearing NaN).
>
> In Eq.(2), we only use $x_i$ as the anchor feature, without using $x_{i\prime}$ as the anchor feature. That is to say, there will be no contrast between the augmented image and abnormal data in our loss. You can simply think about the fact that if $x_{i\prime}$ is not used as the anchor feature, there will be no negative pairs formed between $x_{i\prime}$ and abnormal features. Thus, there's no need to worry about the issue you mentioned.
>
> (2) In line 197, we mentioned that the previous work FCDD [23] stated that the pseudo-Huber distance is a more robust distance measure that interpolates from quadratic to linear penalization; we follow to use this distance. Mathematically, when the value $x$ decreases from large to small, the curve of the pseudo-Huber function will change from quadratic to linear, while the curve of the Euclidean function is always quadratic. When $x$ is small, linear function can provide better gradient information compared to quadratic function. The term $e^{d_i}$ is a weighting factor. In line 205 to 209, we provided a good explanation. When $d_i$ is large (i.e., features are far from the hypersphere), $e^{d_i}$ can provide more weighting. When features are closer to the origin (i.e., $d_i \rightarrow -r$),  $e^{d_i}$ can provide less weighting. Thus, the loss in Eq.(3) can adaptively assign larger gradients to the features outside the hypersphere for better contraction.
>
> (3) In Fig.2, we provided a clear diagram of the Feature Projector, it's not hard to understand when combined with the diagram. The Feature Projector is actually cross-attention combined with MLP, but we use learnable reference representations as Key and Value.
>
> In line 333 to 340, based on Tab.2(b), we provided a clear explanation. In Tab.2(b), for Cross Attention, we convert normal reference features into residual features and then utilize them as Key and Value. Compared to Self Attention, Cross Attention can perform better. Thus, we think that only embodying normal patterns in Key and Value should be effective. Then, we further adopt learnable Key and Value. Compared to limited normal patterns in Cross Attention, the learnable reference can adaptively learn to represent normal patterns and better cover key normal feature patterns. For downstream datasets, it is fixed to ensure the Feature Projector can properly generate good pretrained features. We attempted to set it as learnable, but the features yielded by the Feature Projector will appear pattern collapse issue.
>
> (4) For all losses, $x$ means residual features. In Sec.3.2 and 3.3, we use $x$ as it is commonly used to represent features. In the revision, we will add a note to avoid confusion.
>
> **[To W3]** (1) In line 311 to 313, we stated that all ablation studies were based on the VisA dataset. Thus, the test data comes from the VisA dataset.
>
> (2) **I respectfully argue that our method is not method-dependent, and you may misunderstand the meaning of embedding-based AD methods in our paper**. In the introduction, we explained that we use embedding-based AD methods to refer to AD methods that use pretrained feature networks. So, the embedding-based AD methods in our paper actually cover a wide range. Like the pretrained ResNet-50 on ImageNet, our pretrained features can be applied to any AD methods that use pretrained feature networks. Perhaps this term ''embedding-based'' caused the misunderstanding. In the revision, we will make better modifications for this.
>
> In Tab.1, we utilized 5 AD methods, including PaDiM and PatchCore (feature comparison based), CFLOW (normalizing flow based), GLASS (anomaly synthesis based), and UniAD (reconstruction based), to comprehensively validate the effectiveness of our method. This also demonstrates that our method is not method-dependent. In ablation studies, we evaluate the results based on PaDiM and PatchCore. We also explained in line 269 to 270: these two methods are based on feature comparison without learnable parameters. Thus, the results of these two methods directly depend on the quality of the representation. The better the discriminability between normal and abnormal features, the better the results. Thus, these two methods can directly reflect the quality of pretrained features. For hyperparameter experiments, it's costly for us to evaluate the results on all 5 AD methods, so we think that using PaDiM and PatchCore is the most appropriate.
>
> (3) We think that our contrastive losses are not very sensitive to the three loss hyperparameters. As we found that different loss hyperparameters all can effectively reduce the loss value and converge to a close loss value. This indicates that the optimization effect of the model after loss convergence is similar, so its performance is also close. Compared to the loss hyperparameters, the effects of $N_r$ (see the following (4)) and $K$ are more obvious. In Table 10, the results of Rand(1, 4, 8) and 1 are 94.5/89.3 v.s. 93.8/88.1 under FeatureNorm.
>
> (4) Thanks for your suggestion. We further conducted experiments with $N_r$ set to 32, 64, and 128, and the results are as follows.
>
> |Method|32|64|128|256|2048|2560|3072|4096|
> |-|-|-|-|-|-|-|-|-|
> |PaDiM|92.9/86.4|93.7/87.3|95.2/87.9|95.4/88.1|95.5/88.0|95.5/87.9|95.2/87.6|94.4/87.1|
> |PatchCore|92.1/83.5|93.4/85.2|94.5/86.7|94.8/87.3|94.7/88.0|94.6/87.3|94.4/86.8|93.4/85.6|
> |FeatureNorm|92.0/85.0|92.9/86.5|93.6/87.6|94.0/88.1|93.8/88.1|93.7/88.1|93.4/87.7|92.7/86.4|
>
> It can be seen that the results significantly decrease with $N_r$ set to 32 and 64. This may be because too few learnable references cannot effectively cover key normal feature patterns. We also further conducted experiments with $N_r$ set to 3072 and 4096. When $N_r$ further increases, the results begin to decrease, and an obvious decrease occurs when $N_r$ is 4096. This may be because too many learnable references can lead to difficulties in optimization during training.
>
> **[To W4]** (1) Due to too many experiments (please see Tab.1, with 6 AD methods, 5 datasets, 4 backbones, and with/without our pretrained features, there is a total of $6 \times 5 \times 4 \times 2 = 240$ experiments), we didn't have enough time and resources to use different random seeds for statistical significance. However, we ensured that all experiments were based on the random seed 42. Thus, we eliminated the influence of randomness, ensuring that all result comparisons are fair. In the revision, we will conduct multiple experiments to report the average values and standard deviation.
>
> **We hope the above responses can solve your concerns. If not, we sincerely hope to receive your reply and are very glad to further discuss with you**.
>
> **Note: We try our best to compress the characters, but we still cannot reply to all questions in this box. We borrow some space from the rebuttal box (lower part) of Reviewer rLun to reply to the remaining questions. We sincerely hope that you can take a look**.

---

> > ### Author Response · Authors · 2025-08-04
> >
> > Dear Reviewer oLCH, if you still have any concerns, we sincerely hope to receive your reply and are very glad to further discuss with you.

---

> > ### Comment · Reviewer_oLCH · 2025-08-08
> >
> > I acknowledge the authors’ thoughtful response. The rebuttal has addressed some of my concerns, especially the presented results regarding $N_r$ and the conventional AD methods. I suggest that authors clarify the key experimental settings of the new results to facilitate a clear assessment of the effectiveness of the new results during this or future rebuttal, such as the $N_r$ for conventional AD methods, and key hyperparameters from OCSVM and KNN. Please ensure all new results and supporting statements are incorporated into the revised manuscript. I decide to upgrade my score.

---

> > > ### Author Response · Authors · 2025-08-08
> > >
> > > Thank you for your valuable feedback and for raising the score. The $N_r$ for conventional AD methods is also 2048 to keep consistent with the value in the paper. In KNN, we follow SPADE to set K to 2. In OCSVM, we use the "rbf" kernel and set $\nu$ to 0.5. We will carefully incorporate the results and discussions into the revised paper.

---

### Official Review · Reviewer_SPbR · 2025-06-30

**Clarity:** 2
**Significance:** 4
**Originality:** 3
**Rating:** 5
**Confidence:** 5

**Summary:**

This paper proposes a new contrastive learning strategy and feature adaptation network focusing on industrial anomaly detection scenarios. The ability of residual features is fully exploited, and Learnable Key/Value Attention is proposed for feature adaptation.
Then, Angle-Oriented Contrastive Loss and Norm-Oriented Contrastive Loss are designed to expand the distinction between normal and abnormal features. The sufficient experimental performance improvement on five public datasets proves the advantages of this method. And it can also enhance the adaptability of different types of feature extractors to industrial scenarios.

**Questions:**

See weaknesses.

If the authors could solve my questions, especially weaknesses 2, 4, 7 and 9, I will consider further improving my rating.

**Ethical Concerns:**

["NO or VERY MINOR ethics concerns only"]

**Final Justification:**

After carefully evaluating the authors' responses, I am pleased to confirm that all my concerns have been addressed. While the rebuttals for W3, W5, W8, and W9 do reveal certain limitations in the current work, I recognize this study as a valuable exploration along the path of developing "pretrained industrial anomaly detection models". Considering both the technical contributions and the authors' conscientious rebuttal, I have decided to elevate my rating from Borderline Accept (Rating 4) to Accept (Rating 5).

**Limitations:**

Yes

**Quality:**

3

**Strengths And Weaknesses:**

Strengths:
1. The authors propose two novel contrastive learning losses, Angle-Oriented Contrastive Loss and Norm-Oriented Contrastive Loss, and the theoretical analysis of the two losses is very thorough.  The ablation studies and t-SNE prove that these two losses could increase the difference between normal and abnormal features, thereby improving the performance of the current anomaly detection model.
2. Although residual features are not proposed for the first time in this paper, the analysis and application of residual features in this paper are thorough.  Moreover, the paper designs an effective Learnable Key/Value Attention to transform this feature for subsequent representation learning.
3. The authors conduct sufficient experiments, proving that this training strategy could bring improvements for different AD methods and feature extractors of different architectures.
4. The issue focused on by this paper has great significance to the industrial field. This work could fill the gap in open-source work on feature adaptation in the industrial AD tasks.

Weaknesses:
1. The ablation studies in Table 2 indicate that the non-fixed feature extractor leads to significant performance degradation. This phenomenon means that the weights obtained through pre-training in natural scenarios are very important. This work only adapts the original pre-trained features to industrial scenarios, which is difficult to be defined as "pretrain".
2. line73: "this is the first study dedicated to anomaly representation pertaining." However, the paper **spot-the-difference**[1] proposes a special data augmentation strategy on natural images and uses contrastive learning for pre-training for subsequent industrial anomaly detection. Although the method proposed in this paper is quite different from spot-the-difference, it is beneficial to discuss it in the related works.
3. Why is only the GLASS-h variant used for evaluation? Is it because the Manifold Hypothesis in GLASS-m contradicts the Hypersphere-based loss in this paper?
4. The products in the Real-IAD dataset are all "object classes". Therefore, providing some qualitative and quantitative results on the "texture classes" (such as carpet, tile and grid on the MVTec AD dataset) will be beneficial for better evaluating the generalization performance of features.
5. line 188: "we can downsample the mask to get feature labels." What is the specific downsampling operation (max pooling or average pooling or others) ? Different downsampling strategies could lead to some patches being assigned different labels.
6. Are the output features of the Feature Projector used to calculate both the Angle-Oriented Contrastive Loss and the Regul-oriented Contrastive Loss?  The symbols in this paper are not consistent and clear.
7. In the Learnable Key/Value Attention, the "addition operation" are replaced to "subtraction operation". This improvement requires more explanations and motivations. It seems like performing a secondary residual calculation on the original residual features, but the original residual features have already highlighted the abnormal regions.
8. In the inference phrase, some normal samples need to be provided for calculating residual features, which might limit the wider application of this framework, such as zero-shot and contaminated AD tasks without labeled normal data.
9. In few-shot (especially 2-shot) settings, the influence of reference samples on residual features needs to be discussed. For example, for the screw class of the MVTec AD dataset, samples with different rotation directions are adopted as reference samples. Or, reference samples could be selected through different random seeds for sensitivity analysis.

Some extended problems:
1. Whether the features extracted through the framework in this paper will eliminate the original pre-trained model's ability to distinguish different types of anomalies.
2. Whether it can be applied to supervised anomaly detection tasks by adding a classification head or a segmentation head, such as the experiments in the paper "Spot-the-difference". Although abnormal images are difficult to collect, in real industrial scenarios, supervised anomaly detection models still have very wide applications at present.
The responses to these extended questions will not lead to my negative evaluation of this paper. This paper has provided enough contribution to the AD community.

[1] Zou Y, Jeong J, Pemula L, et al.  Spot-the-difference self-supervised pre-training for anomaly detection and segmentation[C]//European Conference on Computer Vision.  Cham: Springer Nature Switzerland, 2022: 392-408.

---

> ### Author Rebuttal · Authors · 2025-07-28
>
> **[To W1]** We think that the reasons for the performance degradation in the non-fixed backbone may be: (1) the basic visual representation capabilities possessed in the pretrained backbone are still valuable and important, and training the whole feature network on AD datasets may cause impairment to the basic visual representation capabilities. (2) The current scale and quality of AD datasets are not enough to support the whole network training of an AD backbone. The use of the word ''pretrain'' has two reasons: (1) We are indeed following the basic paradigm of pretraining: learning on a large-scale dataset, and the learned features can be transferred to downstream datasets and models. (2) More importantly, we want to emphasize that in the current development stage of anomaly detection, pretrained representations specifically learned for anomaly detection are of great significance.  If it is revised to “pretrained features finetuning”, the significance of this research direction cannot be well reflected.
>
> **[To W2]** In the paper spot-the-difference (SPD), the authors propose a data augmentation strategy called SmoothBlend to create slight local differences (or called pseudo anomalies). Then, the authors create pseudo anomalies on natural images from ImageNet to use as negative samples in contrastive learning. The model is pretrained on ImageNet, while our model is pretrained on a real AD dataset, RealIAD. (1) Compared to pseudo anomalies, the negative samples used in our method are more realistic and more in line with the abnormal characteristics in real-world scenarios. (2) In SPD, similar to classical SSL methods (*SimCLR*, *MoCo*), contrastive loss is constructed based on global features, while we construct contrastive loss based on more fine-grained local features. Although SPD mentioned that it can promote local sensitivity compared to *SimCLR*, *MoCo*, etc., it still may lead to the loss of information about local pseudo anomalies in the global features, as the local augmentation is slight. For fine-grained task like anomaly detection, contrastive learning based on fine-grained local features should be more suitable. (3) SPD was evaluated only on the ResNet-50, PaDiM, and VisA and MVTecAD datasets, while we evaluate our method on more backbones, AD models, and AD datasets.
>
> As the SPD paper was accompanied by the introduction of the VisA dataset, and subsequent works mainly cited this paper due to the VisA dataset, we inadvertently overlooked the method. We greatly appreciate your suggestion and will add a discussion with SPD in the revision.
>
> **[To W3]** Yes, at that time, we mainly felt that the hypersphere hypothesis in GLASS-h was more compatible with our norm-oriented contrastive loss. In the revision, we will also attempt to conduct experiments based on GLASS-m.
>
> **[To W4]** Thanks for your suggestion. In the following two tables, we report the results (image-level AUROC/PRO) on carpet, grid, leather, tile, and wood classes from MVTecAD. The first table is based on CLIP-large, and the second table is based on ImageBind.
>
> | Classes | PaDiM | PaDiM$^†$ | PatchCore | PatchCore$^†$ |
> | - | - | - | - | - |
> | Carpet | 93.9/96.2 | 100+6.1/96.6+0.4 | 98.5/90.7 | 100+1.5/91.7+1.0|
> | Grid | 96.7/91.1 | 99.2+2.5/92.5+1.4 | 91.1/75.7 | 98.6+7.5/79.5+3.8|
> | Leather | 98.4/97.2 | 100+1.6/97.8+0.6 | 100/96.0 | 100+0.0/95.5-0.5|
> | tile | 93.4/84.5 | 99.6+6.2/85.1+0.6 | 99.3/78.4 | 99.7+0.4/81.9+3.5|
> | wood | 98.2/88.7 | 99.5+1.3/91.0+2.3 | 96.9/76.7 | 99.7+2.8/79.8+3.1 |
>
>
> |Classes|PaDiM|PaDiM$^†$|PatchCore|PatchCore$^†$|
> |-|-|-|-|-|
> | Carpet | 100/96.4 | 100+0.0/97.2+0.8 | 100/94.0 | 100+0.0/93.8-0.2|
> |Grid|98.5/93.0|99.7+1.2/93.6+0.6 | 98.8/87.9 | 100+1.2/89.4+1.5|
> |Leather | 99.9/97.5 | 100+0.1/97.7+0.2 | 100/96.1 | 100+1.6/96.7+0.6|
> |tile | 98.8/85.6 | 99.1+0.3/85.1-0.5 | 98.2/83.0 | 98.4+0.2/82.6-0.4|
> |wood | 99.1/92.3 | 99.5+0.4/92.7+0.4 | 98.1/85.2 | 99.1+1.0/86.2+1.0|
>
> It can be found that the results on texture classes can also be effectively improved. We are sorry that we cannot provide you with qualitative results directly, as the rebuttal text box does not support uploading images. Previously, NeurIPS supported submitting a one-page PDF file to upload images, but this function was canceled this year. In the revision, we will add the corresponding qualitative results.
>
> **[To W5]** For the feature map with a downsampling ratio of $s$, we divide the original mask into patches with $s\times s$ size. Then, for each patch, we sum the number of 1s (1 represents abnormal in the ground-truth mask). If the number of 1s is larger than 32, we label this patch as 1. Compared to ''nearest'' downsampling, our downsampling method can preserve more patches labeled as 1. Because the number of abnormal patches is much smaller than that of normal patches, we found that the ''nearest'' downsampling can cause instability in the angle-oriented contrastive loss (*i.e.*, NaN).
>
> **[To W6]** Yes, the features yielded by the Feature Projector are used to calculate both the Angle-Oriented Contrastive Loss and the Norm-Oriented Contrastive Loss.
>
> **[To W7]** In line 333 to 340, we provided some explanations to our Learnable Key/Value Attention based on the results in Tab.2(b). In Tab.2(b), for Cross Attention, we convert normal reference features into residual features and then utilize them as Key and Value. Compared to Self Attention, Cross Attention can perform better. Thus, we think that only embodying normal patterns in Key and Value should be effective. Then, we further adopt learnable Key and Value. Compared to limited normal patterns in Cross Attention, the learnable reference should can adaptively learn to represent normal patterns and better cover key normal feature patterns in the residual feature distribution during pretraining. By subtraction, we aim to eliminate the normal representations in the residual feature distribution adaptively learned by the network, further increasing the discrepancy between normal and abnormal residual features.
>
> **[To W8]** The need for some normal samples is a limitation mainly from residual features. However, we think that collecting a small amount of uncontaminated normal samples is usually feasible and easy for practical use. From the essence of anomalies (defined relative to normal), we think few-shot is more reasonable than zero-shot because at least we need to know what is normal and then detect anomalies. About contamination AD, you can see our response to Weakness 2 of Reviewer rLun, we provide some results and a discussion.
>
> **[To W9]** We greatly appreciate your suggestion. Residual features are mainly relied on mutually eliminating class-related normal patterns in features through subtraction (you could see the ResAD paper for more explanations), for being class-generalizable. However, if normal images have significant variability, it may cause the few-shot reference samples to be not representative enough (lacking some class-related normal patterns). This may lead to certain class-related normal patterns in the input feature can't be effectively eliminated. Then, the discriminability of residual features under few-shot settings will be compromised.
>
> For practical applications, this issue should be particularly focused and reasonably addressed. Of course, the simplest solution is to increase the number of reference samples. This is feasible, as in practical applications, the number of reference samples is usually not as strict as the 2-shot. However, in practical applications, we expect that the reference samples can fully represent their class, so it’s best to have sufficient differences between the reference samples. Thus, the sample selection strategy cannot be random. A feasible method is to first cluster all available normal samples into different clusters based on a clustering algorithm (e.g., KMeans). Then, based on the number of reference samples, we evenly distribute it to each cluster. When selecting from a cluster, we can prioritize selecting samples closer to the center. Another feasible strategy is to match spatially aligned samples as reference samples for each input sample. This will be helpful for the example you mentioned, as we can match samples with similar rotation directions to the object in the input sample as reference samples. We have provided a corresponding method in Appendix B.
>
> In the revision, we will add the above discussion about the influence of reference samples on residual features and the sample selection strategy to the paper.
>
> **[To Q1]** Yes, because one characteristic of residual features is that the semantic discriminability in features will be eliminated. However, for anomaly detection, it is okay not to distinguish different types of anomalies, and most AD methods can only detect anomalies and cannot give anomaly types.
>
>
> **[To Q2]** We think that our pretrained features can be used for supervised anomaly detection tasks. In supervised AD tasks, the model is more likely to learn good decision boundaries between normal and abnormal based on our pretrained features by effectively utilizing the abnormal samples during training. Because our pretrained features have better discriminability for representing normal and abnormal compared to the original features. Due to time limitation, we currently don't have enough time to provide the corresponding experimental results. In the revision, we will attempt to apply our pretrained features to supervised AD methods.
>
> **We hope the above responses can solve your concerns. If not, we sincerely hope to receive your reply and are very glad to further discuss with you**.

---

> > ### Comment · Reviewer_SPbR · 2025-08-03
> >
> > I appreciate the authors' responses to most of my concerns and their commitment to addressing these issues in the revision.
> >
> > 1. Their rebuttal regarding W1 unconvincing. The approach of attaching lightweight "adapters" to a fixed pretrained network does not constitute true "pretraining" as claimed. The method proposed in this paper is conceptually the same as methods like April-GAN and other CLIP-based zero-shot approaches, where linear layers are added for feature adaptation after the fixed CLIP model. The current implementation appears to be a form of feature adaptation rather than pretraining per se.
> > 2. Regarding W5: The authors' rebuttal states that "our downsampling method can preserve more patches labeled as 1." This claim naturally raises the question: would max-pooling potentially yield better performance for mask downsampling compared to the current approach?
> > 3. The motivation behind the "subtraction operation" remains unclear in the authors' response about W7. While they mention that "we aim to eliminate the normal representations in the residual feature distribution adaptively learned by the network, further increasing the discrepancy between normal and abnormal residual features," this explanation requires stronger empirical support. Specifically, Visual evidence (e.g., t-SNE maps) would be ideal to demonstrate the claimed effects, though we understand such analysis may be challenging during the rebuttal period.
> > At a minimum, an ablation study on the subtraction operation should be conducted to quantify its impact on performance metrics. We strongly recommend that the authors include more rigorous experimental validation in their revision to verify whether it indeed enlarges the gap between normal/anomalous features as claimed.

---

> > > ### Author Response · Authors · 2025-08-03
> > >
> > > We greatly appreciate your careful responses to us. In the following, we will provide further explanations to the three comments in your reply.
> > >
> > > **[To R1]** We acknowledge that, from the implementation perspective, our method is not true "pretraining". Due to the reasons mentioned in previous **[To W1]**, our method didn't achieve pretraining the whole backbone network. So, you should think that using the word “pretraining” is not rigorous enough. However, we think that the "anomaly representation pretraining" (learning pretrained representations specifically for AD tasks) is a very valuable research direction for anomaly detection, and our work is only an early exploration. We hope to inspire more future works to focus on anomaly representation pretraining. Therefore, based on your suggestion, in the revised paper, we will only use the word "pretraining" in the introduction of the "anomaly representation pretraining" task. For our method, we will revise to "AD representation adaptation framework" or other suitable words.
> > >
> > > **[To R2]** Thanks for your further suggestion. It does not mean that the more patches labeled as 1, the better the performance. In the "max-pooling", if the label of one pixel is 1, the corresponding patch will be labeled as 1. We think that this is not very reasonable. Thus, we didn't utilize the "max-pooling" downsampling but employed the threshold-based downsampling strategy (only when the number of 1s is larger than the threshold, the patch will be labeled as 1). With the "max-pooling" downsampling, we have also conducted an experiment (based on PatchCore and ImageBind). The results on VisA are 94.0/87.8 (v.s. 94.7/88.2). As we felt that the downsampling strategy is not a very important part, we didn't specifically conduct ablation studies. In the revision, we will conduct more comprehensive experiments for this.
> > >
> > > **[To R3]** When writing the paper, we have conducted corresponding experiments. But in Tab.2(b), we mainly compare our Learnable Key/Value Attention with other network structures, so we didn't include the ablation study about the "subtraction operation". The corresponding ablation study is as follows:
> > >
> > > | | PaDiM | PatchCore | FeatureNorm |
> > > | - | - | - | -|
> > > |addition operation | 95.2/88.6 | 94.4/86.8 | 94.0/88.2 |
> > > |subtraction operation | 95.5/88.8 | 94.7/87.6 | 94.5/89.3 |
> > >
> > > For the visual evidence, we appreciate that you understand that we cannot directly provide such analysis in the text box. We will include the corresponding visualization figures in the revision.
> > >
> > > **We hope the above responses can solve your concerns. If you still have any concerns, we are very glad to further discuss with you**.

---

### Official Review · Reviewer_rLun · 2025-07-02

**Clarity:** 4
**Significance:** 4
**Originality:** 3
**Rating:** 4
**Confidence:** 5

**Summary:**

This paper addresses the intriguing and under-explored topic of anomaly representation pretraining, a critical yet scarcely researched area in unsupervised anomaly detection. Existing state-of-the-art AD methods heavily rely on ImageNet-pretrained features, which suffer from mismatched objectives (e.g., classification vs. anomaly discrimination) and distribution shifts between natural and industrial images. The authors propose ADPretrain, a novel framework that leverages the large-scale RealIAD dataset to learn robust pretrained representations by integrating residual features and designing angle- and norm-oriented contrastive losses. This approach aims to maximize discriminative differences between normal and abnormal features while mitigating distribution shifts.  While the study acknowledges limitations such as reliance on embedding-based AD methods and fixed backbone networks, its focus on task-specific pretraining fills a significant research gap. By pioneering anomaly-focused pretraining, the work offers valuable insights and paves the way for future advancements, making it a promising contribution despite its room for improvement.

**Questions:**

- Compare ADPretrain with fine-tuning methods (e.g., FYD, PANDA) on the same datasets to quantify the benefit of task-specific pretraining.
- Introduce synthetic noise or corrupted samples in experiments to assess robustness.
- Validate the framework on emerging backbones (EfficientNet, DINOv2) to confirm its applicability across modern architectures.

**Ethical Concerns:**

["NO or VERY MINOR ethics concerns only"]

**Final Justification:**

After several rounds of discussion, the author has addressed most of my questions, so I have decided to increase the score somewhat. I have a positive attitude toward the paper, but since the overall analysis is still not particularly impressive, my overall rating has not increased.

**Limitations:**

yes

**Quality:**

3

**Strengths And Weaknesses:**

### **Strengths**
- The paper addresses the limitations of using ImageNet-pretrained features for anomaly detection by proposing a task-specific pretraining framework, with well-designed residual features and contrastive losses tailored for AD.

- The related work section provides a thorough review of existing methods and clearly positions the novelty of the proposed approach.


### **Weaknesses**
-  While the paper emphasizes that ADPretrain outperforms ImageNet-pretrained features, it does not explicitly compare its framework with state-of-the-art fine-tuning methods like FYD or mean-shifted contrastive loss. Such a comparison is crucial to validate whether pretraining truly offers advantages over adaptive fine-tuning strategies in addressing distribution shifts .

-  The experiments do not test the framework under more settings, such as noisy conditions where normal samples might be corrupted. For example, methods like SoftPatch [1] have shown robustness to noise, but ADPretrain’s reliance on residual features—derived from clean normal references—may struggle if training data contains noise. This gap leaves uncertainty about its performance in real-world scenarios with imperfect data.

-  It does not evaluate modern architectures like EfficientNet or DINOv2, which have demonstrated superior representation capabilities in self-supervised learning. Additionally, while the paper mentions that residual features are class-generalizable, it does not explicitly validate their effectiveness on cutting-edge backbones (e.g., UniFormaly’s findings on ViT vs. CNN [2]). Expanding to more architectures would strengthen the generalizability claim.

[1] Jiang X, Liu J, Wang J, et al. Softpatch: Unsupervised anomaly detection with noisy data[J]. Advances in Neural Information Processing Systems, 2022, 35: 15433-15445.

[2] Lee Y, Lim H, Jang S, et al. Uniformaly: Towards task-agnostic unified framework for visual anomaly detection[J]. Pattern Recognition, 2025: 111820.

---

> ### Author Rebuttal · Authors · 2025-07-29
>
> **Dear Reviewer rLun, due to time limitation and to respond to multiple reviewers, we currently don’t have enough time to provide experiment results as comprehensive as in the paper. In the revision, we will supplement the results on other datasets**.
>
> **[To W1]** As the FYD paper does not provide open-source code, we can only compare with the results provided in the paper. In FYD, anomaly detection is achieved based on PaDiM. The paper provides the results (with WideResNet50) on MVTecAD, which are 97.7/98.2 (image-level/pixel-level AUROCs). Based on WideResNet50 and PaDiM, the results of our method are 98.1/98.5. In PANDA and mean-shift contrastive loss (MSC), feature adaptation is based on global features, so only image-level anomaly detection is performed. The two papers provide image-level AUROC on MVTecAD, with values of 86.5 and 87.2. For a more reasonable comparison, we utilized the open-source code of MSC and trained an adapted network (based on WideResNet50) on MVTecAD. Then, we utilized PaDiM for AD modeling, and the obtained results are 96.8/97.0. Thus, compared with PANDA, MSC, and FYD, our method has advantages in providing better features for AD models. As discussed in Related Work, adaptive feature fine-tuning requires specialized fine-tuning on each dataset. By comparison, a more important advantage of anomaly representation pretraining is that pretrained features can be used on multiple downstream AD datasets without the need for fine-tuning on each dataset.
>
> **[To W2]** Thanks for your suggestion. We further conducted experiments with noisy data following the settings in SoftPatch. Specifically, we added abnormal data from the test set to the training set with a noise ratio of 0.1. We adopted the ''overlap'' setting from the SoftPatch paper, which means the injected abnormal images are also included in the test set. The results are as follows (based on ImageBind, image-level AUROC/PRO).
>
> |Datasets|PaDiM|PaDiM$^†$|PatchCore|PatchCore$^†$|
> |-|-|-|-|-|
> |MVTecAD|86.6/82.7|88.2$\textcolor{green}{+1.6}$/84.6$\textcolor{green}{+1.9}$|87.7/80.6|89.2$\textcolor{green}{+1.5}$/81.4$\textcolor{green}{+0.8}$|
> |VisA|82.8/75.6|87.4$\textcolor{green}{+4.6}$/79.9$\textcolor{green}{+4.3}$|83.4/71.5|86.9$\textcolor{green}{+3.5}$/78.7$\textcolor{green}{+7.2}$|
>
> The results without noisy data are as follows.
>
> |Datasets|PaDiM|PaDiM$^†$|PatchCore|PatchCore$^†$|
> |-|-|-|-|-|
> |MVTecAD|98.3/93.0|98.9$\textcolor{green}{+0.6}$/93.2$\textcolor{green}{+0.2}$|98.4/89.8|98.8$\textcolor{green}{+0.4}$/89.8$\textcolor{gray}{+0.0}$|
> |VisA|92.6/86.3|95.6$\textcolor{green}{+3.0}$/88.6$\textcolor{green}{+2.3}$|91.6/81.3|94.7$\textcolor{green}{+3.1}$/88.2$\textcolor{green}{+6.9}$|
>
> It can be seen that the performance of the AD models significantly decreases when there is noisy data. Although the performance of the AD models with our pretrained features also decreases (this is normal, our paper is not intended to tackle the contamination AD task). However, our pretrained features can bring greater performance improvement under noisy data. This indicates that our pretrained features are more robust to noise than the original ImageNet-pretrained features. Moreover, our method is compatible with contamination AD methods, such as SoftPatch. When the training data has noise, we can first utilize the filtering method in SoftPatch to remove the most likely abnormal data, and then train based on the remaining data.
>
> Finally, we note that for the above results, we don't add abnormal data to the normal reference samples. This is because when generating residual features, we only used few-shot and fixed normal samples as reference (i.e., 8 samples). We think that in real-world scenarios, even if training data has noise, it's easy and almost costless to manually collect few-shot normal samples as reference. Thus, we think that it seems not necessary to forcefully introduce abnormal data to the few-shot normal reference samples. In the revision, we will add the above results and discussion and cite the paper you mentioned.
>
> **[To W3]** Thanks for your suggestion. Based on your suggestion, we further conducted a set of experiments based on EfficientNet-b6 and DINOv2-large, and the results are as follows (image-level AUROC/PRO).
>
> |Model|Datasets|PaDiM|PaDiM$^†$|PatchCore|PatchCore$^†$|CFLOW|CFLOW$^†$|GLASS|GLASS$^†$|UniAD|UniAD$^†$|
> |-|-|-|-|-|-|-|-|-|-|-|-|
> |EfficientNet-b6|MVTecAD|95.3/87.8|96.8$\textcolor{green}{+1.5}$/88.5$\textcolor{green}{+0.7}$|98.6/92.0|98.9$\textcolor{green}{+0.3}$/92.5$\textcolor{green}{+0.5}$|97.9/91.6|98.5$\textcolor{green}{+0.6}$/92.4$\textcolor{green}{+0.8}$|98.7/93.6|99.0$\textcolor{green}{+0.3}$/94.7$\textcolor{green}{+1.1}$|96.9/89.5|97.7$\textcolor{green}{+0.8}$/90.7$\textcolor{green}{+1.2}$|
> | |VisA|86.4/76.7|90.2$\textcolor{green}{+3.8}$/84.3$\textcolor{green}{+7.6}$|93.5/87.7|94.0$\textcolor{green}{+0.5}$/88.3$\textcolor{green}{+0.6}$|93.3/86.1|93.7$\textcolor{green}{+0.4}$/86.8$\textcolor{green}{+0.7}$|94.2/88.6|94.7$\textcolor{green}{+0.5}$/89.6$\textcolor{green}{+1.0}$|92.7/85.9|94.3$\textcolor{green}{+1.6}$/87.2$\textcolor{green}{+1.3}$|
> |DINOv2-large|MVTecAD|98.7/91.0|98.6$\textcolor{red}{-0.1}$/92.4$\textcolor{green}{+1.4}$|97.8/84.9|98.0$\textcolor{green}{+0.2}$/84.6$\textcolor{red}{-0.3}$|98.8/92.7|98.9$\textcolor{green}{+0.1}$/93.2$\textcolor{green}{+0.5}$|98.4/95.3|99.1$\textcolor{green}{+0.7}$/96.2$\textcolor{green}{+0.9}$|96.6/89.3|97.1$\textcolor{green}{+0.5}$/89.6$\textcolor{green}{+0.3}$|
> | |VisA|92.6/85.6|95.1$\textcolor{green}{+2.5}$/86.7$\textcolor{green}{+1.1}$|84.2/71.4|85.9$\textcolor{green}{+1.7}$/75.8$\textcolor{green}{+4.4}$|96.2/90.0|96.9$\textcolor{green}{+0.7}$/90.6$\textcolor{green}{+0.6}$|93.3/90.4|94.0$\textcolor{green}{+0.7}$/91.8$\textcolor{green}{+1.4}$|87.6/85.8|90.0$\textcolor{green}{+2.4}$/86.9$\textcolor{green}{+1.1}$|
>
> It can be seen that we can also achieve performance improvement on EfficientNet-b6 and DINOv2-large. In the revision, we will supplement the results on other datasets and cite the paper you mentioned.
>
> **[To Q1]** Please see our response to Weakness 1.
>
> **[To Q2]** Please see our response to Weakness 2.
>
> **[To Q3]** Please see our response to Weakness 3.
>
> **We hope the above responses can solve your concerns. If not, we sincerely hope to receive your reply and are very glad to further discuss with you**.
> ***
> ***
> ***
> ***
> ***
> ***
> ***
> ***
> ***
> ***
> **Note: [To Reviewer rLun], you don't need to see the following content**. Due to space limitation, we borrow some space to answer some questions of Reviewer oLCH. We sincerely apologize for any inconvenience caused to you.
>
> **[To W4]** (2) As stated in Abstract, we promise that we will open-source the code, it's not to worry. When submitting the paper, we didn't organize all code well (with sufficient instructions), so we were unable to directly include it in the supplementary materials.
>
> **[To Q1]** In the Norm-Oriented Contrastive Loss, the basic idea follows one-class-classification (OCC) learning. In OCC learning, normal features are usually optimized to be located inside the unit hypersphere (i.e., when $||x_i|^2_2 = 1$, it locates on the unit hypersphere). In the pseudo-Huber distance $\sqrt{||x_i|^2_2+1}-1$, when $||x_i|^2_2 = 1$,  $\sqrt{||x_i|^2_2+1}-1 \approx 0.4$. Setting $r$ to 0.4 is equivalent to the unit hypersphere based on Euclidean distance.
>
> **[To Q2]** WideResNet50 outputs two layers of features, with dimensions of 512 and 1024. The features generated by CLIP-base/CLIP-large/ImageBind are all 768/1024/1280 dimensions. The feature dimension has no direct relation to detection performance. Detection performance mainly depends on the feature network. It's not that with a larger feature dimension, the detection performance must be better.
>
> **[To Q3]** In line 274 to 275, we mentioned that "we further propose a simple AD baseline (denoted by FeatureNorm), where we directly utilize feature norms as anomaly scores". For few-shot AD, our FeatureNorm is really simple. Specifically, for an input sample, we extract pretrained residual features from our pretrained model. Then, for a pretrained residual feature $x_i$, the L2 norm $||x_i||_2$ of $x_i$ is used as the anomaly score. In the revision, we will add this more specific description to the paper.
>
> **[To L1]** We further conducted experiments based on OCSVM and KNN, and the results are as follows (based on ImageBind).
>
> |Datasets|OCVSM|OCVSM$^†$|KNN|KNN$^†$|
> |-|-|-|-|-|
> |MVTecAD|84.8/77.9|90.4$\textcolor{green}{+5.6}$/82.8$\textcolor{green}{+4.9}$|92.6/84.7|97.8$\textcolor{green}{+5.2}$/88.4$\textcolor{green}{+3.7}$|
> |VisA|75.3/72.2|81.4$\textcolor{green}{+6.1}$/77.3$\textcolor{green}{+5.1}$|80.4/75.6|87.5$\textcolor{green}{+7.1}$/80.2$\textcolor{green}{+4.6}$|
>
> It can be seen that our pretrained features can achieve significant performance improvement, which further indicates that our pretrained features have better discriminability between normal and abnormal. Due to time limitation, we only conducted experiments on two datasets. In the revision, we will supplement experiments on other datasets and try other algorithms.
>
> **[To L2]** In Tab.2(b), we have conducted ablation studies about MLP (the MLP Projector in Tab.2(b)). Does the ''readout layer'' refer to the network layers that output features? We further conducted experiments, and the results are as follows.
>
> |Architecture|PaDiM|PatchCore|FeatureNorm|
> |-|-|-|-|
> |Output Layer Learnable|94.0/87.9|93.8/86.5|93.2/85.7|
> |Learnable Key/Value Attention (ours)|95.5/88.8|94.7/87.6|94.5/89.3|
>
> Compared to training the whole network, only training the output layers can significantly improve the results. This may be because fixing most parameters can effectively preserve the basic visual representation capabilities possessed in the pretrained backbone, which are still valuable and helpful. We greatly appreciate your suggestion. In the revision, we will include these ablation studies you mentioned.

---

> > ### Author Response · Authors · 2025-08-04
> >
> > Dear Reviewer rLun, if you still have any concerns, we sincerely hope to receive your reply and are very glad to further discuss with you.

---

> > ### Comment · Reviewer_rLun · 2025-08-05
> >
> > Thank you for your detailed rebuttal and the additional experiments. I appreciate your efforts to address my comments. However, I still have several concerns and questions regarding your responses:
> >
> > 1. [To W1]:
> >    I feel that your reply does not fully resolve my concerns. A comparison based solely on performance metrics is insufficient. It is crucial to clarify that FYD does not utilize large-scale anomaly detection datasets for pretraining when improving feature representations. What impact does this have on feature fine-tuning? This is a fundamental difference that should be discussed in depth, as it directly affects the generalizability and transferability of the learned features. I think you may need to provide more analysis and visualization to illustrate this point. (Reviewer SPbR also raised a similar concern.)
> >
> > 2. [To W2]:
> >    I have further questions regarding your noise robustness experiments. In your setup, are the reference samples fixed? My original intention was to discuss the impact of noise specifically in the reference samples on the overall performance. Your algorithm does not necessarily need to be robust to all types of noise (which could be the focus of future work), but theoretically, if the pretrained model has a stronger representation of anomalies, it might actually be more sensitive to noise—especially in methods like PatchCore. However, your results show improved robustness, which is counterintuitive and raises further questions. Could you clarify this phenomenon and provide more analysis?
> >
> > 3. Additional Question after Re-reading the Paper:
> >    Regarding Figure 3, are the visualized features from the pretrained model, or are they obtained after supervised training on the VisA dataset? The visual separation in the visualization is very pronounced, which suggests that even a simple classifier could achieve very high performance. Please clarify the source of these features and whether the visualization reflects the true unsupervised setting.
> >
> > Comments on Other Reviewers’ Opinions:
> >
> > - I agree that your method aligns well with the traditional definition of "unsupervised industrial anomaly detection" (this term itself can be somewhat misleading).
> > - I find the use of the term "pretraining" appropriate and helpful for understanding the purpose of your work, and I support keeping it.

---

> > > ### Author Response · Authors · 2025-08-06
> > >
> > > We greatly appreciate your careful responses to us. In the following, we will provide further explanations to the three comments in your reply.
> > >
> > > **[To R1** We think that the scale of the pretraining dataset will have an impact, a larger scale is more advantageous for pretraining (better transferability of the learned features). This is also in line with common practice: pretraining is usually based on a large-scale dataset. In our response to Reviewer yUBc's Q2, we discussed this point and provided some experimental results. We performed pretraining on the MVTecAD dataset, and the results are overall lower than the results obtained based on RealIAD pretraining. The corresponding results are as follows (based on ImageBind, Image-level AUROC/PRO):
> > >
> > > | Datasets | PaDiM(RealIAD) | PaDiM(MVTecAD) | PatchCore(RealIAD) | PatchCore(MVTecAD) |
> > > |-|-|-|-|-|
> > > | MVTecAD | 98.9/93.2 | 97.7/92.5 | 98.8/89.8 | 98.8/88.0 |
> > > | VisA | 95.6/88.6 | 93.0/85.6 | 94.7/88.2 | 93.4/86.0 |
> > > | BTAD | 95.9/76.8 | 95.6/76.6 | 95.6/67.3 | 94.7/66.5 |
> > > | MVTec3D | 84.4/92.0 | 82.8/91.2 | 82.6/87.0 | 80.6/87.3 |
> > > | MPDD | 94.4/95.1 | 92.9/93.5 | 94.8/94.2 | 93.1/93.4 |
> > >
> > > In the FYD paper, the transferability of the fine-tuned features is not discussed. In FYD, after fine-tuning on MVTecAD, the fine-tuned model is effective on MVTecAD, but it has not been verified whether it is still effective on other datasets. We think that FYD requires specialized fine-tuning on each dataset. The reason is: The fine-tuning process in FYD is mainly to accomplish spatial alignment (beneficial to the PaDiM method) on feature maps. As the spatial distributions of objects in different datasets are different, the learned affine transformation parameters in one dataset should be hard to apply to other datasets. In our paper, the results in Tab.1 have comprehensively verified that the pretrained features have good transferability in downstream AD datasets.
> > >
> > > For visualization, we are sorry that we cannot directly provide you with visualization figures, as the text box does not support uploading images. In the revision, we will add the above discussion and the results, and also provide corresponding qualitative results.
> > >
> > > **[To R2]** Yes, we use 8 fixed normal samples as reference. For the robustness, we think that a possible explanation may be: Although some noises are added to the training set, it is still dominated by normal features. In PatchCore, the training process is to subsample a coreset. With better representation, abnormal features are more likely to be sparse outliers, making them more likely not to be sampled into the coreset. Thus, the results show better robustness. For the impact of noise in the reference samples, we further conducted an experiment by adding one abnormal sample (the first abnormal sample in each class) to the reference samples. Then, the reference set contains 7 normal samples and 1 abnormal sample. The results are as follows.
> > >
> > > | Datasets | PaDiM (w/o noise) | PaDiM (with noise) | PatchCore (w/o noise) | PatchCore (with noise) |
> > > |-|-|-|-|-|
> > > | MVTecAD | 98.9/93.2 | 98.0/92.0 | 98.8/89.8 | 98.2/88.5 |
> > > | VisA | 95.6/88.6 | 93.8/86.3 | 94.7/88.2 | 93.1/85.6 |
> > >
> > > The presence of noise in the reference samples can lead to performance degradation. This is normal because residual features are generated by matching the closest reference features and subtracting. If there are abnormal patterns in the reference features, this will cause these abnormal patterns to be matched to some abnormal region features. Then, the generated residual features will be closer to normal, causing unnecessary confusion. However, we think that noise in the reference samples is not a very serious issue, because in real-world scenarios, even if training data has noise, it's easy and almost costless to manually collect few-shot (e.g., 8) normal samples to construct a clean reference set.
> > >
> > > **[To R3]** The visualized features are from the pretrained model. For downstream AD datasets, we only used normal samples for AD modeling and didn't use abnormal samples for further supervised training. Thus, all experiments are unsupervised on downstream AD datasets. For Figure 3, to make the visualization more intuitive and clear, we select some best-performing samples for visualization. If this is not very reasonable, we will modify this to use all sample features for visualization in the revision.
> > >
> > > **We hope the above responses can solve your concerns. If you still have any concerns, we are very glad to further discuss with you**.

---

### Official Review · Reviewer_yUBc · 2025-07-05

**Clarity:** 2
**Significance:** 2
**Originality:** 2
**Rating:** 4
**Confidence:** 2

**Summary:**

This paper introduces ADPretrain, an anomaly-aware pretraining framework designed to generate better feature representations for unsupervised anomaly detection. The authors propose a pseudo-anomaly discrimination task along with multi-scale feature fusion to make the pretraining more aligned with downstream AD tasks.

**Questions:**

Could the authors compare ADPretrain against other pretrained representations such as MAE, DINO, or industrial-specific SSL models?

What is the impact of pretraining dataset choice on generalization? Is ADPretrain truly domain-agnostic?

Given the add-on nature of ADPretrain, is it possible to integrate it into fully end-to-end AD pipelines (e.g., UniAD), rather than only use it for feature extraction?

Could the authors elaborate on whether their method helps in resource-constrained scenarios (e.g., faster inference, fewer training images)?

**Ethical Concerns:**

["NO or VERY MINOR ethics concerns only"]

**Final Justification:**

The authors clarified the method’s novelty and added useful experiments supporting its effectiveness and generalization. While I am not an expert in this specific subfield, the response seems reasonable and improves my overall impression of the work. I am therefore raising my score from 3 to 4.

**Limitations:**

The authors have clearly acknowledged the main limitations of their work, including the fixed backbone design and the restricted applicability to embedding-based AD methods. These reflections are fair and sufficient given the scope of the paper

**Paper Formatting Concerns:**

The paper is well-formatted and adheres to the NeurIPS style guidelines.

**Quality:**

3

**Strengths And Weaknesses:**

Strengths

The key strengths of this paper is that it tackles a limitation in the current AD landscape.
The results are good. Across a wide variety of datasets and multiple backbones, the method improves performance.

Weaknesses

1.	The paper lacks direct comparisons against alternative pretrained backbones beyond ImageNet (e.g., MAE, DINOv2, self-distillation), which makes it difficult to contextualize the performance gain solely to the proposed pretraining strategy.
2.	The method essentially acts as a drop-in feature encoder, and while the performance improvements are notable, the core contribution is mostly centered around representation learning rather than architectural novelty.
3.	The paper would benefit from further analysis of computational cost, especially regarding whether ADPretrain can reduce downstream training/inference cost or improve sample efficiency.

---

> ### Author Rebuttal · Authors · 2025-07-29
>
> **Dear Reviewer yUBc, due to time limitation and to respond to multiple reviewers, we currently don’t have enough time to provide experiment results as comprehensive as in the paper. In the revision, we will supplement the results on other datasets**.
>
> **[To W1]** Thanks for your professional review. We use the phrase ''ImageNet-pretrained'' not to mean that the backbones used in experiments are all from ImageNet pretraining, but to refer to visual feature networks pretrained on all sorts of large-scale datasets (not just ImageNet). Because ImageNet is the most well-known pretraining dataset, we thus still use ''ImageNet-pretrained''. We are sorry that this caused misunderstanding and will modify the phrase in the revision. The CLIP and ImageBind models (please see Table 1) used in experiments are based on self-supervised learning (i.e., contrastive language-image pretraining), and are not pretrained on ImageNet. Our experimental comparison way is to separately use the original features and our pretrained AD representations in an AD model (in Table 1, we conducted experiments on 5 AD models), train the AD model, obtain the results, and then compare the corresponding results based on the two features. We keep all other aspects of the model consistent, so the differences in results only stem from the different features. Therefore, the comparison way can directly reflect that our pretrained features are better than the original features when used in AD models.
>
> Based on your suggestion, we further conducted a set of experiments based on DINOv2-large, and the results are as follows.
>
> | Model | Datasets | PaDiM | PaDiM$^†$ | PatchCore | PatchCore$^†$ | CFLOW | CFLOW$^†$ | GLASS | GLASS$^†$ | UniAD | UniAD$^†$|
> |-|-|-|-|-|-|-|-|-|-|-|-|
> |DINOv2-large | MVTecAD | 98.7/91.0 | 98.6$\textcolor{red}{-0.1}$/92.4$\textcolor{green}{+1.4}$ | 97.8/84.9 | 98.0$\textcolor{green}{+0.2}$/84.6$\textcolor{red}{-0.3}$ | 98.8/92.7 | 98.9$\textcolor{green}{+0.1}$/93.2$\textcolor{green}{+0.5}$ | 98.4/95.3 | 99.1$\textcolor{green}{+0.7}$/96.2$\textcolor{green}{+0.9}$ | 96.6/89.3 | 97.1$\textcolor{green}{+0.5}$/89.6$\textcolor{green}{+0.3}$|
> | | VisA  | 92.6/85.6 | 95.1$\textcolor{green}{+2.5}$/86.7$\textcolor{green}{+1.1}$ | 84.2/71.4 | 85.9$\textcolor{green}{+1.7}$/75.8$\textcolor{green}{+4.4}$ | 96.2/90.0 | 96.9$\textcolor{green}{+0.7}$/90.6$\textcolor{green}{+0.6}$ | 93.3/90.4 | 94.0$\textcolor{green}{+0.7}$/91.8$\textcolor{green}{+1.4}$ | 87.6/85.8 | 90.0$\textcolor{green}{+2.4}$/86.9$\textcolor{green}{+1.1}$|
>
> The results show that we can also achieve performance improvement based on DINOv2-large. Because there are a large number of experiments under one backbone, we currently don't have enough time to run all the backbones you mentioned. In the revision, we will complete the results and validate our pretraining method on other backbones you mentioned. In addition, please see our response to Weakness 3 of Reviewer rLun. We also provide the experiment results based on EfficientNet-b6.
>
> **[To W2]** We think that in the current development stage of anomaly detection, fundamental representation is more important than architectural novelty. In previous stages, many papers have proposed various architectural novelties, but overlooked that the essence of anomaly detection still can be attributed to the representation quality of features. In fact, architectural novelty is also aimed at better learning how to distinguish between normal and abnormal. If the normal and abnormal features are already highly discriminative (e.g., linearly separable), anomaly detection tasks can be easily accomplished without the need for designing increasingly complex architectural novelties. We also observe that most of the architectural novelties in AD papers are specific to the proposed method and are often hard to be still effective in other methods. However, good representations will be effective in various AD methods.
>
> The current mainstream AD models mostly use pretrained networks to extract features. However, regardless of supervised or self-supervised pretraining on natural images, the pretraining process does not match the goal of anomaly detection. From the perspective of the development prospects of anomaly detection, we think that it's necessary for AD tasks to have specialized pretrained features instead of continuously using basic pretrained features. We think that pretrained representations specifically learned for anomaly detection are of great significance, and our work is only an early exploration. We hope to inspire more future works to focus on anomaly representation pretraining.
>
> **[To W3]** When our pretrained features are used in downstream AD models, feature extraction and specific AD modeling are still required. Thus, there is no reduction in inference cost. We further conducted an experiment (only 10\% of the normal samples from each dataset are used for training), and obtained the following results (based on ImageBind).
>
> | Datasets | PaDiM | PaDiM$^†$ | PatchCore | PatchCore$^†$ |
> |-|-|-|-|-|
> | MVTecAD | 81.4/88.6 | 96.8$\textcolor{green}{+15.4}$/90.3$\textcolor{green}{+1.7}$ | 96.5/86.2 | 98.2$\textcolor{green}{+1.7}$/87.7$\textcolor{green}{+1.5}$ |
> | VisA | 82.8/79.3 | 93.0$\textcolor{green}{+10.2}$/83.3$\textcolor{green}{+4.0}$ | 88.9/78.7 | 93.9$\textcolor{green}{+5.0}$/85.7$\textcolor{green}{+7.0}$|
>
> The results based on full training data are as follows.
>
> | Datasets | PaDiM | PaDiM$^†$ | PatchCore | PatchCore$^†$ |
> |-|-|-|-|-|
> | MVTecAD | 98.3/93.0 | 98.9$\textcolor{green}{+0.6}$/93.2$\textcolor{green}{+0.2}$ | 98.4/89.8 | 98.8$\textcolor{green}{+0.4}$/89.8$\textcolor{gray}{+0.0}$ |
> | VisA | 92.6/86.3 | 95.6$\textcolor{green}{+3.0}$/88.6$\textcolor{green}{+2.3}$ | 91.6/81.3 | 94.7$\textcolor{green}{+3.1}$/88.2$\textcolor{green}{+6.9}$ |
>
> The results show that with less training data, our pretrained features can bring more significant performance improvement, indicating that our pretrained features are beneficial for the sample efficiency (during training) of downstream AD methods.
>
> **[To Q1]** Please see our response to Weakness 1.
>
> **[To Q2]** As pretraining is usually based on a large-scale dataset, and currently most AD datasets only have a few thousand samples, we thus choose RealIAD for pretraining due to its data scale. To answer your question, we further performed pretraining on the MVTecAD dataset. The obtained results are as follows (based on ImageBind).
>
> | Datasets | PaDiM(RealIAD) | PaDiM(MVTecAD) | PatchCore(RealIAD) | PatchCore(MVTecAD) |
> |-|-|-|-|-|
> | MVTecAD | 98.9/93.2 | 97.7/92.5 | 98.8/89.8 | 98.8/88.0 |
> | VisA | 95.6/88.6 | 93.0/85.6 | 94.7/88.2 | 93.4/86.0 |
> | BTAD | 95.9/76.8 | 95.6/76.6 | 95.6/67.3 | 94.7/66.5 |
> | MVTec3D | 84.4/92.0 | 82.8/91.2 | 82.6/87.0 | 80.6/87.3 |
> | MPDD | 94.4/95.1 | 92.9/93.5 | 94.8/94.2 | 93.1/93.4 |
>
> It can be seen that with pretraining on MVTecAD, the results are overall lower than the results obtained based on RealIAD pretraining. Thus, we think that the scale of the dataset will have an impact, a larger scale is more advantageous for pretraining. This also reflects that the AD field still needs larger and higher-quality datasets in the future to better support anomaly representation pretraining.
>
> We are not quite sure about your definition of domain-agnostic. Generally, domain-agnostic means that a trained model can also perform well in other domains or datasets. From this perspective, our pretrained features can achieve better results than the original features when used on 5 downstream datasets and 5 AD methods. This indicates that our pretrained features are domain-agnostic (at least better than the original features) and method-agnostic. If there is any deviation between our understanding of domain-agnostic and yours, we hope to further discuss with you in the subsequent discussion phase.
>
> **[To Q3]** Do you mean to combine our method with UniAD and train it on the RealIAD dataset? We think that residual features and the Feature Projector could be integrated into UniAD. These two contrastive losses could not be applied to UniAD because they are designed for representation learning, while the learning objective in UniAD is reconstruction. But we think that even if UniAD can perform well after training on RealIAD, it is only a specific AD model and cannot empower other AD models, while the pretrained fundamental representations are more valuable and can be used to enhance downstream AD models.
>
> **[To Q4]** Please see our response to Weakness 3.
>
> **We hope the above responses can solve your concerns. If not, we sincerely hope to receive your reply and are very glad to further discuss with you**.

---

> > ### Author Response · Authors · 2025-08-04
> >
> > Dear Reviewer yUBc, if you still have any concerns, we sincerely hope to receive your reply and are very glad to further discuss with you.

---

> > ### Comment · Reviewer_yUBc · 2025-08-08
> >
> > The rebuttal adequately addresses my main concerns. The authors clarified the method’s novelty and added useful experiments supporting its effectiveness and generalization. While I am not an expert in this specific subfield, the response seems reasonable and improves my overall impression of the work. I am therefore raising my score from 3 to 4.

---

> > > ### Author Response · Authors · 2025-08-08
> > >
> > > Thank you for your valuable feedback and for raising the score. We will carefully incorporate the corresponding results and discussions into the revised paper.

---

### Decision · Program_Chairs · 2025-09-17

**Decision:**

Accept (poster)

**Comment:**

This paper proposes a pre-training strategy for unsupervised abnormality detection. To this end, the authors proposed to pre-train the backbones with labeled AD datasets with angle- and norm-oriented objectives over residual features, then swaps these representations into standard unsupervised AD methods. Across various datasets and multiple backbones, the swapped-in features consistently improve detection/segmentation metrics.

The paper received generally positive reviews with two borderline accepts, one accept, and one borderline reject. The critical concerns raised by the reviewers include:

- **Misleading terminology:** Describing the overall pipeline as “unsupervised” obscures that pre-training employs the labeled data.
- **Fairness of evaluation:** to show the improvement is not backbone- or recipe-specific, the paper should systematically cover strong modern pretrained reps/backbones (e.g., MAE/DINOv2, EfficientNet/ViT) and report variance/error bars to support reproducibility.
- **Comparison to fine-tuning baselines.** A controlled comparison against per-dataset fine-tuning methods (e.g., FYD, mean-shift contrastive, PANDA) is needed to justify that plug-in representations offer advantages beyond what standard fine-tuning already delivers.
- **Robustness and reference-set sensitivity.** Since residual features depend on few-shot normal references, behavior under contaminated normals and noisy/misaligned reference sets must be analyzed, with a causal explanation for why relative gains persist (or fail) across noise regimes.

During the rebuttal, the authors addressed many of these concerns reasonably, leading some reviewers to increase their scores. While not discussed in the rebuttal, the AC believes the paper misses an important ablation: because both pre-training and evaluation lie in industrial AD, gains could be a trivial consequence of a small domain gap or data overlap. The paper should therefore demonstrate pretrain–test dissimilarity (i.e., no leakage) rather than assume it. This analysis of the pre-training and test data, together with the additional experiments and clarifications presented in the rebuttal, **must** be included in the final version.